# Abrupt reduction in shipping emission as an inadvertent geoengineering termination shock produces substantial radiative warming
Tianle Yuan [1,2] ✉, Hua Song [2,3], Lazaros Oreopoulos [2], Robert Wood [4], Huisheng Bian[1,2], Katherine Breen[2,5], Mian Chin [2], Hongbin Yu[2], Donifan Barahona[2], Kerry Meyer[2] & Steven Platnick[2]

Human activities affect the Earth's climate through modifying the composition of the atmosphere, which then creates radiative forcing that drives climate change. The warming effect of anthropogenic greenhouse gases has been partially balanced by the cooling effect of anthropogenic aerosols. In 2020, fuel regulations abruptly reduced the emission of sulfur dioxide from international shipping by about 80% and created an inadvertent geoengineering termination shock with global impact. Here we estimate the regulation leads to a radiative forcing of $+0.2 \pm 0.11 \, \mathrm{Wm}^{-2}$ averaged over the global ocean. The amount of radiative forcing could lead to a doubling (or more) of the warming rate in the 2020 s compared with the rate since 1980 with strong spatiotemporal heterogeneity. The warming effect is consistent with the recent observed strong warming in 2023 and expected to make the 2020 s anomalously warm. The forcing is equivalent in magnitude to 80% of the measured increase in planetary heat uptake since 2020. The radiative forcing also has strong hemispheric contrast, which has important implications for precipitation pattern changes. Our result suggests marine cloud brightening may be a viable geoengineering method in temporarily cooling the climate that has its unique challenges due to inherent spatiotemporal heterogeneity.

The Earth's atmosphere has warmed because of human activities increasing the concentration of greenhouse gasses that trap thermal radiative energy in the climate system, creating a positive climate forcing. Human activities have also increased the concentration of aerosol particles that can affect the amount of reflected solar radiation back to space either directly or indirectly by interacting with clouds, which has an overall cooling effect on the climate[1]. The magnitude of the aerosol cooling effect has significant implications for estimating how sensitive our climate is to greenhouse gas forcing and the amount of expected future warming for a given increase of greenhouse gas concentrations[2]. The effectiveness of anthropogenic aerosols in cooling the climate also has direct implications for solar radiation modification geoengineering schemes[3,4]. Such methods aim to produce temporary cooling of the climate through enhanced reflection of solar radiation to space. They are not solutions to greenhouse gas induced global warming and have uncertain and complex additional consequences besides the intended short-term cooling effect[4–7].

Marine cloud brightening (MCB) is a type of solar radiation modification scheme where marine low clouds are seeded with aerosols to become brighter[8,9]. Examples of small scale, opportunistic MCB experiments were discovered in early satellite observations of ship-tracks, linear features of brighter oceanic clouds because of ship-emitted aerosols[10,11]. The addition of aerosols from ship emissions results in more cloud droplets, leading to more reflective clouds for a given amount of total In-cloud liquid water, or liquid water path (LWP)[12]. More recent studies show that aerosols can also change LWP and total cloud fraction (CF), which also greatly affect the amount of solar radiation reflected by clouds[2,13–16].

Aerosols sourced from global shipping industry affect clouds and we can view the shipping emission as a long-running inadvertent MCB

[1]GESTAR-II, University of Maryland, Baltimore County, MD, USA. [2]Sciences and Exploration Directorate, Goddard Space Flight Center, Greenbelt, MD, USA. [3]SSAI, Inc., Lanham, MD, USA. [4]Department of Atmospheric Sciences, University of Washington, Seattle, WA, USA. [5]GESTAR-II, Morgan State University, Baltimore, MD, USA. ✉e-mail: tianle.yuan@nasa.gov

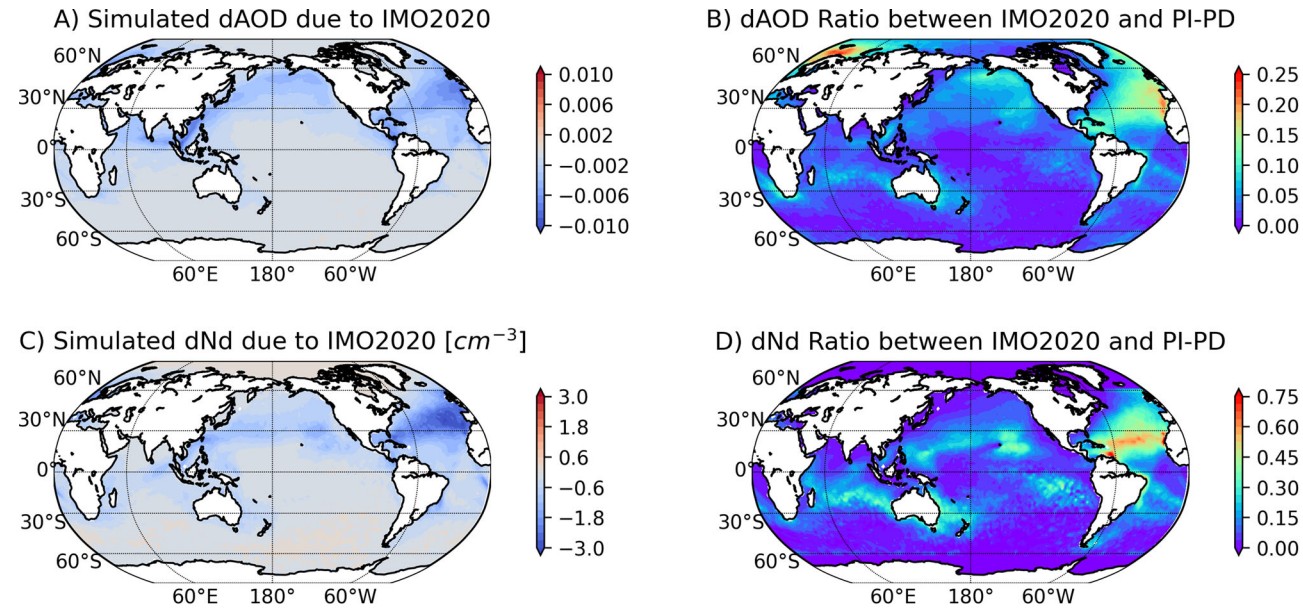

**Fig. 1 | Simulated impact of IMO2020 on AOD and $N_d$. A** simulated annual mean aerosol optical depth change induced by IMO2020 using NASA GOES-GOCART. **B** the ratio of aerosol optical depth changes between that induced by IMO2020 and

that between 1750 and 2005[2]. **C** map of simulated annual mean $N_d$ change due to IMO2020. **D**) same as **B**, but for $N_d$ change.

experiment. On January 1, 2020, new International Maritime Organization (IMO) regulations on the sulfur content of international shipping fuel took effect. The IMO 2020 regulation (IMO2020) reduced the maximum sulfur content from 3.5% to 0.5%[17]. While IMO2020 is intended to benefit public health by decreasing aerosol loading, this decrease in aerosols can temporarily accelerate global warming by dimming clouds across the global oceans. IMO2020 took effect in a short period of time and likely has global impact. IMO2020 effectively represents a termination shock for the inadvertent geoengineering experiment through a reverse MCB, i.e., marine cloud dimming through reducing cloud droplet number concentration ($N_d$) (Fig. 1). Observations of ship-tracks suggest that IMO2020 has reduced the occurrence and modified the properties of ship-tracks across global oceans, demonstrating that a regulation intended to reduce pollution had collateral effects on cloud microphysics[18]. Analyses of remote sensing data have shown evidence of cloud dimming in the South Atlantic shipping lane[19]. Outside the South Atlantic, the effect of IMO2020 does not have a distinct spatial structure[18,19], which makes direct observation of the impact more challenging.

## Results

Here we combine satellite observations and a chemical transport model to quantify the radiative forcing of the inadvertent geoengineering event induced by IMO2020 and estimate its climate impacts. We simulate the impact of IMO2020 on maritime aerosol concentrations with the NASA GEOS-GOCART model. Figure 1 shows the modeled reduction in aerosol optical depth due to decreased $SO_2$ emission from the international shipping industry. The AOD reduction reaches peak values of around 0.01 in the South China Sea and Eastern North Atlantic off the coasts of Western Europe. In the South Atlantic the regulations create AOD reductions that follow the shape of shipping routes. We then calculate the ratio between the AOD change due to IMO2020 and that between pre-industrial and present day. Over most of the ocean, the ratio is smaller than 10% because of sparse shipping outside the major shipping routes. Over the North Pacific and North Atlantic, on the other hand, it can exceed 10% and reaches 25% in the Norwegian Sea and off the western European and northwestern African coasts. In these regions, the total anthropogenic aerosol concentration is relatively low because of declining emissions of aerosols and their precursors since the 1980s, making ship-emitted aerosols an important component of

the anthropogenic maritime aerosols. The IMO2020 is therefore effective in reducing total aerosol loading for these regions. The impact of IMO2020 on the cloud droplet number concentration ($N_d$) of low-level maritime clouds as shown in Fig. 1C (see Methods). Globally, IMO2020 leads to a modest reduction of 0.5 $cm^{-3}$ in mean modeled $N_d$. Regionally, however, the reduction is more pronounced. The strongest reduction occurs in the North Atlantic, the Caribbeans and the South China Sea, reaching 3 $cm^{-3}$. These are regions with the busiest shipping lanes and thus strongest reduction of ship emissions. The reduction in the South Atlantic shows the most well-defined shipping lane shapes likely due to the unique circulation pattern in this region[18,20]. Figure 1D shows the ratio between IMO2020 induced Nd decrease and estimated Nd difference between preindustrial and present day. The ratio is small over the major outflow areas downwind of major continental sources, but becomes substantially larger in more remote oceans, reaching 30%. In the tropical North Atlantic, IMO2020 induced change in Nd can be more than 50% of the total anthropogenic change.

We combine $N_d$ changes due to IMO2020 with satellite observations to estimate the forcing introduced by the inadvertent geoengineering event[21]. We consider both the Twomey effect and the effects of cloud liquid water path (LWP) and cloud fraction adjustments to $N_d$ (see Methods section). The LWP and cloud fraction adjustments follow the functional forms derived from a large sample of ship-tracks[21] that depend on the background cloud $N_d$, sea surface temperature (SST), estimated inversion strength, and background low cloud fraction (see Methods). Figure 2 shows the pattern of annual mean of forcing resulting from $N_d$ decrease due to IMO2020 averaged over different LWP and cloud fraction adjustment functional forms. The total forcing is $+0.2 \pm 0.11\,\mathrm{Wm^{-2}}$ averaged over the global ocean with the Twomey effect contributing 40%, the LWP adjustment being near neutral, and the cloud fraction adjustment contributing 60%. The positive radiative forcing has strong regional variations. The North Atlantic experiences the strongest radiative forcing peaking around 1.4 $\mathrm{Wm^{-2}}$ and whose basin-wide mean is around 0.56 $\mathrm{Wm^{-2}}$. Weaker but still notable radiative forcing is seen in the North Pacific and the South Atlantic. This ordering is consistent with the amount of ship traffic and low cloud fraction in these regions. Our estimate of radiative forcing from IMO2020 is well within the range of estimates of the total forcing from shipping emissions in the literature[22–25]. We also compare our estimate with that from a recent observational study in the core shipping lane in the South Atlantic that used

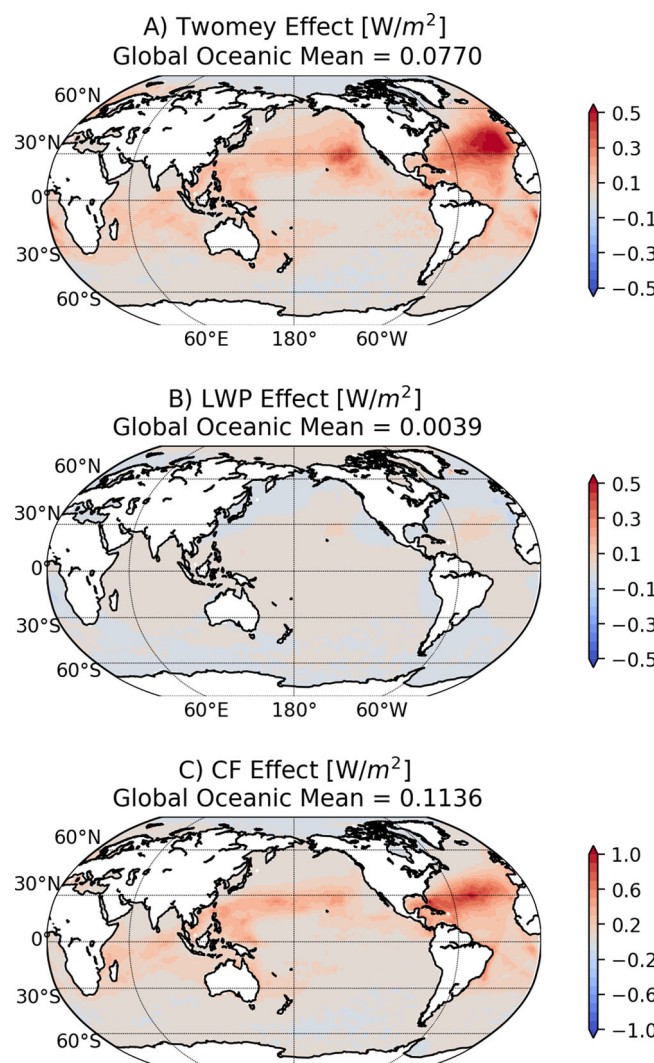

**Fig. 2 | Calculated IMO2020 forcing maps from different components.** The spatial patterns of three components of forcing from cloud adjustments: **A** the Twomey effect, **B** LWP adjustment, and **C** cloud fraction adjustment.

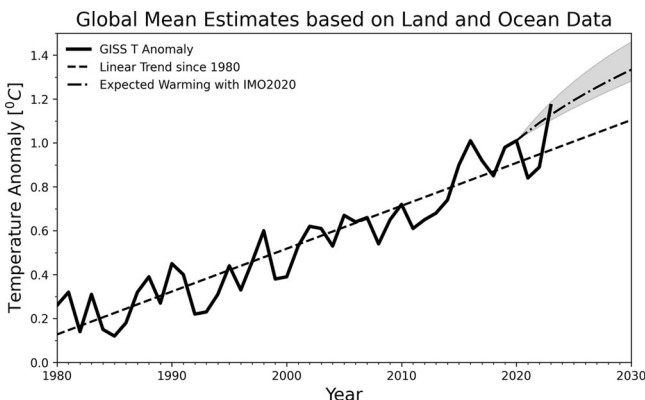

**Fig. 3 | Time series of global temperature anomaly since 1980 (Lensen et al., 2019).** The trend line is dashed. The expected warming trajectory from the combination of the linear trend and the calculated warming effect from IMO 2020 shock based on the energy balance model. The upper and lower bounds of the expected warming are shown in shades. The baseline period for temperature anomaly is between 1951 and 1980.

a different approach[19]. The two completely independent approaches yield very similar radiative forcing in the core shipping lane (supporting online material, SOM), which serves as a cross-validation. Similar global forcing, i.e., on the order of 0.1Wm$^{-2}$, is reported by multiple modeling groups[26].

Using an energy balance model[27], we calculate the expected amount of transient temperature increase due to warming resulting from IMO2020. For simplicity, we ignore the heat uptake by the deep ocean during the short-term, i.e. O(10) years. 0.2 W m$^{-2}$ translates to around 0.16 K of warming with a timescale of 7 years. It is equivalent to 0.24 K/decade, which is more than double the average warming rate since 1880 and 20% higher than the mean warming rate since 1980, linear trend of 0.19 K/decade. We also calculated the lower and upper bounds of the forcing and corresponding expected warming (Fig. 3). The IMO2020 is expected to provide a substantial boost to the warming rate of global mean temperature in the 2020 s. The rate of warming is expected to ramp up quickly from 2020 and asymptotes to the longer-term trend line at the end of 2020[27]. The 2023 record warmth is within the ranges of our expected trajectory. The magnitude of IMO2020 induced warming means that the observed strong warming in 2023 will be a new norm in the 2020 s. The mean temperature anomaly of the 2020 s will be 0.3 K higher than that of the 2010s. Regionally, the warming effect from IMO2020 on SST is harder to estimate since basin-wide SST changes can be affected by variations in factors like other aerosol

concentration, ocean circulation, and air-sea interactions. However, the strong geographical variations in the forcing suggest the impact of IMO2020 on SST may have significant variation among ocean basins. For example, the North Atlantic SST may be disproportionally warmed more by the IMO2020 given the radiative forcing is more than three times the global average, which is likely a contributing factor to the pronounced warming of the North Atlantic SST in recent years[28]. A more robust quantitative estimate of the contribution of IMO2020 to regional SST warming requires coupled global climate models that have good representation of aerosol indirect effects.

The IMO2020-induced radiative forcing exhibits considerable seasonal variations. This is evident in the North Atlantic where the IMO2020 produced the strongest forcing. Figure 4 shows the monthly mean time series of forcing and its three components. We use a simple functional form for cloud adjustments that only depends on background $N_d$ to illustrate the point. The total forcing varies between 0.19Wm$^{-2}$ and 0.38Wm$^{-2}$, a 100% relative change. The seasonal variation of incoming solar radiation is the dominant driver for this (SOM). But seasonal variations of background CF, $N_d$, and $\Delta N_d$ due to IMO2020 also contribute as they affect the magnitude of the Twomey effect and macrophysical (LWP and CF) cloud adjustments. We estimate the contribution from each variable after removing the seasonal change in solar insolation, and report the results in Fig. 4B–D (see Methods). $\Delta N_d$ induced by IMO2020 is the strongest contributor. Its variations can affect the forcing by more than 30% in some months such as Jan, Apr, and Dec. Its impact on LWP and CF adjustments contributes equally to the total radiative forcing. The seasonal variation of background $N_d$ is also an important factor (Fig. 4C). Background CF also meaningfully contributes to the seasonal variations through mostly affecting the Twomey effect (Fig. 4D).

We compare the radiative forcing due to IMO2020 and its effect on radiative energy balance with observed changes in relevant quantities. The comparison does not prove causality but provides a context to assess the impact of IMO2020. The low cloud dimming forcing of 0.2 Wm$^{-2}$ from the IMO2020 represents a strong temporary shock to the net planetary heat uptake (Fig. 5A) that has been increasing at a rate of ~0.05 Wm$^{-2}$/yr[29] in measurements. The net planetary heat uptake has increased by 0.25 Wm$^{-2}$ since 2020, making the 0.2 Wm$^{-2}$ due to IMO2020 nearly 80% of the total increase. The long-term trend of CERES TOA net radiation is 0.46 Wm$^{-2}$/decade while it changes to 0.67 Wm$^{-2}$/decade since IMO2020 took effect. The difference is 0.21 Wm$^{-2}$ that is consistent with our estimated forcing. However, the record since 2020 is too short to ascertain the impact of IMO2020 on the long-term trend of the energy balance given its large interannual variations. The IMO2020 effect also has an asymmetric impact

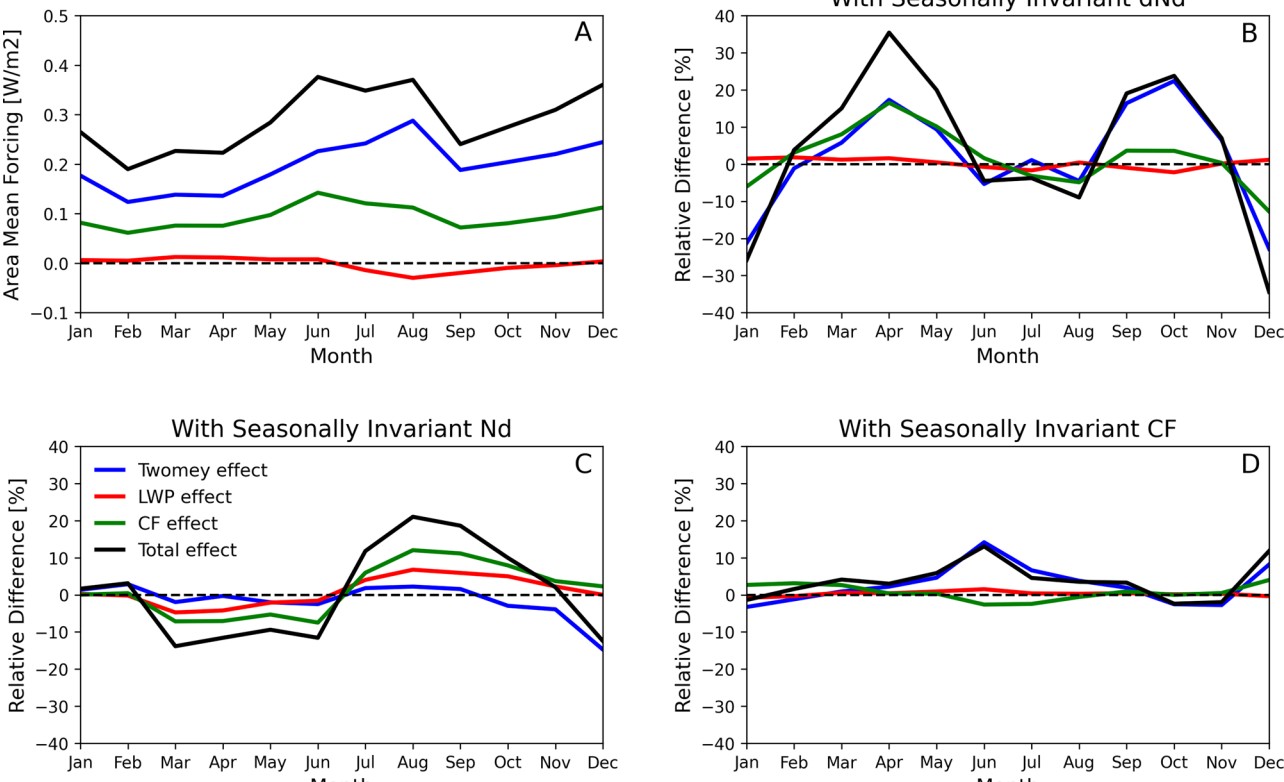

**Fig. 4 | Sensitivity of forcing components to different cloud variables. A** The areal mean of forcing from IMO2020 in the North Atlantic (0°−80°W, 0° ~ 60°N) and its break down in three components. **B–D** sensitivity tests to gauge the impacts of seasonal variations in $\Delta N_d$, background $N_d$, and cloud fraction, respectively. In each test, we use an annual mean map instead of seasonally changing fields to calculate the radiative forcing and plot their difference from the baseline. Details in Methods section.

on aerosol loading in the northern and southern hemispheres because of higher baseline ship emissions in the northern hemisphere. This creates interhemispheric contrast in the resulting radiative forcing, which has important implications for deliberate geoengineering schemes because interhemispheric forcing contrast can create significant perturbations in precipitation patterns[6]. We calculate the interhemispheric contrast in IMO2020 induced warming effect to be around 0.22 Wm$^{-2}$, with the northern hemisphere at 0.32 Wm$^{-2}$ and the southern hemisphere at 0.1 Wm$^{-2}$. The 0.22 Wm$^{-2}$ contrast is substantial when compared with recent measured changes in the interhemispheric contrast in absorbed solar radiation. Figure 5B, C shows measured time series of top-of-atmosphere (TOA) absorbed solar radiation of both hemispheres and their contrast, respectively. Since IMO2020 took effect, the northern hemisphere (NH) absorbed solar radiation has increased by 0.5 Wm$^{-2}$ from a plateau between 2017 and 2020 while the southern hemisphere (SH) increased at a much slower rate. The low cloud dimming effect of IMO2020 represents around 60% of increase in NH absorbed solar radiation. The interhemispheric contrast in absorbed solar radiation has increased by ~0.2 Wm$^{-2}$ based on measurements, one of the highest rates of increase during the whole record, which is almost the same as that induced by low cloud dimming effect of IMO2020. Another rapid increase period is associated with a phase shift in Pacific Decadal Oscillation (PDO) starting 2014/2015 followed by a strong El-Nino event[29]. The PDO entered a negative phase in 2020, which would favor a further decrease in the contrast rather than the observed increase. It is worth noting that in addition to modes of ocean variability such as PDO and El-Nino Southern Oscillation may contribute to variations in these quantities[6].

The combination of modeled $\Delta N_d$ and observed relationship for LWP and cloud fraction adjustments show that IMO2020 as a termination shock for the inadvertent geoengineering experiment of shipping emissions has

had a non-trivial warming effect on the climate. The National Academy of Sciences, Engineering, and Medicines[4] recommended the impact of any outdoor solar radiation management experiment on global mean temperature to be within $1 \times 10^{-7}$K. The forcing magnitude of this inadvertent shock has exceeded this limit by a large margin. However, it does suggest that MCB is a viable solar radiation modification scheme in temporarily slowing the rate of climate warming. Our analysis also points to strong spatiotemporal heterogeneities in the forcing produced by the event. Such heterogeneities need to be considered in any MCB scheme to minimize their potential undesired impacts on regional climate in addition to the desired slowing of climate warming rate. Important part of the heterogeneity exists because of background low cloud distribution and its spatiotemporal variability creating an interhemispheric contrast of radiative forcing having similar magnitude as the global mean radiative forcing. Understanding this contrast is important because to achieve the goal of substantially slowing down the warming rate or limit the maximum warming to be within 1.5[30], much larger forcing than that of IMO2020, but of the opposite sign, would be needed. As a result, the interhemispheric contrast needs to be minimized to avoid substantial perturbations to regional monsoons and other precipitation patterns. It should be noted that the forcing due to IMO2020 will take time for it to be directly detectable at the global scale in the observation records, but regionally, e.g., in the North Atlantic, its impact may be detectable sooner. Regional radiative forcing is already detectable in the Southeast Atlantic shipping lane[19]. Finally, an important open question for policy makers to consider is the trade-off between the benefits of better air quality and the potential cost of additional warming as different parts of the world have reduced and are going to reduce aerosol pollution[31,32]. The trade-off consideration is also relevant for deliberate geoengineering schemes to select the right properties of emitted aerosols.

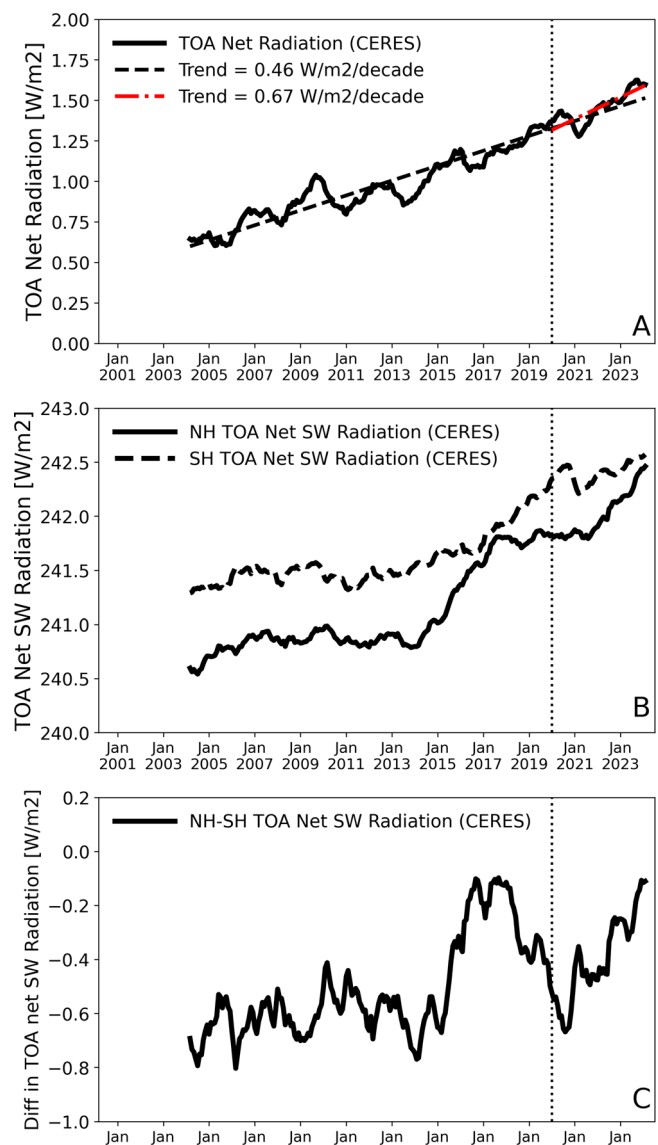

**Fig. 5 | Observed time series of energy balance variables. A** the planetary heat uptake; **B** trailing 48-month mean of absorbed solar radiation for both hemispheres. The 48-month mean is applied to remove high-frequency noise. **C** Time series of Interhemispheric contrast in absorbed solar radiation. The vertical dotted line marks the Jan 2020—details in the Methods section.

There are several sources of uncertainties in our estimate of the radiative forcing via cloud dimming induced by IMO2020. A key source is the magnitude of $N_d$ change. Here the $N_d$ change is modeled with a chemical transport model and not constrained with actual observations. The annual mean change in $N_d$ ($0.5\,cm^{-3}$) is small compared to the background $N_d$ ($28\,cm^{-3}$) and its variability. Counterfactual analyses of satellite-based $N_d$ changes due to ship emissions in the South Atlantic may provide useful regional constraint on $\Delta N_d$ once there are additional years of observations[19]. Although adjustments of LWP and cloud fraction are robust given the large number of samples, they have their own limitations as detailed in previous studies[14,33–36]. One way to gauge the possible range of uncertainty for our forcing estimate is to compare ours with that from Diamond (2023)[19] in the South Atlantic (SOM). In the core shipping lane, the forcing is estimated to peak around $0.5\,Wm^{-2}$, the Twomey effect being the dominant factor in this region (see Fig. 2), in our analysis in excellent agreement with theirs[19]. The inadvertent nature of IMO2020 means that the ratio between radiative

forcing and changes in aerosol mass is not optimized. Here we report $0.2\,Wm^{-2}$ for around 3.7 Tg of S reduction, which is much less efficient than a more optimized scheme due to factors such as emitted aerosol size distribution and the spatial distribution of emission changes[9]. Finally, our analysis does not consider feedback processes. The additional warming of the ocean can induce positive feedbacks from low clouds[37,38], which can only be addressed in a coupled climate model.

In summary, IMO2020 represents a termination shock for the inadvertent geoengineering by global ship emissions through a reverse MCB and produces a positive forcing of $+0.2 \pm 0.11\,Wm^{-2}$. It is expected to provide strong additional warming rate this decade, more than doubling the long-term mean warming rate. The forcing has pronounced spatiotemporal heterogeneity. The IMO2020 effect also contributes to a strong temporary increase to the planetary heat uptake through cloud dimming, and it is around 80% of the measured increase in interhemispheric contrast of absorbed solar radiation since 2020. Our results offer useful guidance for MCB and aerosol-cloud interaction research.

## Methods
### GEOS-GOCART simulation of IMO impact on aerosol fields
All simulation experiments were run with the Goddard Chemistry Aerosol Radiation and Transport (GOCART) aerosol module[39,40] in NASA Goddard Earth Observing System (GEOS) Earth System Model (ESM). The GEOS model has a one-moment cloud microphysics module and a rapid radiation transfer model for general circulation models (RRTMG). Sea surface temperature (SST) for the atmospheric dynamic circulation is provided by the GEOS Atmospheric Data Assimilation System (ADAS) that incorporates satellite and in situ SST observations. The model is run in the replay mode using meteorological fields from the Modern-Era Retrospective Analysis for Research and Applications version 2 (MERRA-2) reanalysis[41]. "Replay" mode sets the model dynamic state (winds, pressure, and temperature) every 6 h to the balanced states provided by the Meteorological Reanalysis Field of the Modern Research and Applications Reanalysis Version 2 (MERRA-2). We run GEOS at a global horizontal resolution of approximately 50 km on a cubic sphere grid and 72 vertical layers from the surface to 0.01 hPa. The time step for dynamic calculation is 450 s. The temporal resolution of the radiation is 1 h. All experiments run from 201910 to 202012, with the first three months as the spin up period. We use monthly results in our estimation of forcing.

We have two set of experiments: business as usual (BAU) and Covid impact (Covid) emissions. The BAU used anthropogenic emissions of aerosols and precursor gases from the Community Emission Data System (CEDS)[42] but repeat the 2019 emissions for 2020. The dataset includes nine emission sectors (energy, industry, road transportation, residential, waste, agriculture, solvent, shipping, and air traffic). Biomass burning emissions were taken from the GSFC-developed Quick Fire Emission Dataset (QFED)[43]. Volcanic emissions come from the dataset that is based on the satellite volcanic SO2 observations from the OMI instrument on board the Aura satellite. Biogenic emissions were calculated with the Model of Emissions of Gases and Aerosols from Nature (MEGAN) that is embedded in GEOS model. The wind-driven emissions, such as dust and sea salt, were calculated on-line. Time varying greenhouse gases, such as $CO_2$, $CH_4$, $N_2O$ and ozone-depleting substances, were provided by CMIP5 project.

The second set of experiments (Covid) adjusted BAU anthropogenic emission to reflect the impact of Covid. Using mobility data from Google and Apple[44], daily scale factors in 2020 were derived on sector bases for ten species including important aerosols and their precursors. Because of the rapidly changing emissions due to various timing and strength of lockdown measures, daily scale factors were provided not only to scale down emission amounts but also to move emissions from monthly to daily. The Covid adjusted daily anthropogenic emissions were generated by applying these scale factors to CEDS 2019 monthly emission.

A summary of S emissions under different scenarios is provided in Table S1. For each set of experiments, there are three scenarios: full ship emissions, reduced ship emissions using the IMO2020 standards, and no

ship emission of S. Other emissions are kept the same. We take the difference in aerosol loading between with full ship emissions of $SO_2$ and with reduction due to IMO2020 as the impact of IMO2020. The difference in aerosol loading is translated into $N_d$ changes with method in the following subsection.

## Deep learning-based $N_d$

The operational version of NASA's Global Earth Observing System (GEOS) runs single moment cloud and aerosol microphysical schemes. They do not predict cloud condensation nuclei (CCN) and cloud droplet number concentrations ($N_d$). We estimate $N_d$ using a diagnostic deep learning-based approach, involving the usage of two neural network (NN) parameterizations. The first NN (termed MAMnet) is an emulator for the Modal Aerosol Module, which takes bulk aerosol mass for 5 externally-mixed species (sulfates, sea salt, dust, black carbon, and organics) and the atmospheric state (temperature, pressure) as input, and predicts the number concentration and composition for 7 internally-mixed lognormal modes (accumulation, aitken, coarse/fine dust, coarse/fine sea salt, primary carbon matter). MAMnet was trained on 5 years of data from a GEOS simulation implementing the MAM7 aerosol module, and validated against ground observations. The CCN concentration at a given supersatureation can be readily estimated using the 7-modal size distribution and composition[45]. A second NN model (Wnet) is used to estimate $N_d$. Estimation of $N_d$ requires the characteristic vertical wind velocity (W) at the scale of individual parcels, typically proportional to its subgrid-scale standard deviation ($\sigma_W$). Wnet takes the atmospheric state, as well as coarse metrics of turbulence (Richardson number and total scalar diffusivity) as inputs and predicts $\sigma_W$ for each grid cell. Wnet was trained on 2 years of a global, non-hydrostatic, storm-resolving simulation of the GEOS model and physically constrained by ground-based observations of $\sigma_W$ from around the world using a novel generative approach[46]. The aerosol size distribution predicted from MAMnet as well as $\sigma_W$ are used to predict $N_d$ using the Abdul-Razzak and Ghan scheme[47]. Our NN-based method emphasizes observations and conservation of known physics during the development of the NNs and ensures a robust prediction of CCN and $N_d$.

## Calculating aerosol indirect forcing

We use the same methodology reported in Yuan et al.[21]. We consider the Twomey effect and effects of cloud fraction and LWP adjustments. Without considering aerosol effects on cloud fractions, cloud albedo sensitivity to aerosols can be taken as the sum of the Twomey effect and aerosol induced LWP adjustments:

$$S = \frac{dA_c}{dN_d} = \frac{A_c(1 - A_c)}{3N_d} \times \left(1 + \frac{5}{2}\frac{dlnLWP}{dlnN_d}\right) \quad (1)$$

where S is the susceptibility of cloud albedo ($A_c$) to droplet number concentration $N_d$[16].

We then have

$$\Delta SW_{TOA} = -SW_{downwelling} \times Cf \times S \times \Delta N_d \quad (2)$$

Aerosol indirect forcing from the Twomey effect and LWP adjustment is therefore:

$$\Delta SW_{TOA} = -SW_{downwelling} \times Cf \times A_c \times (1 - A_c)$$
$$\times \left(\frac{1}{3} + \frac{5}{6}\frac{dlnLWP}{dlnN_d}\right) \times \Delta lnN_d \quad (3)$$

To consider the effect of Cf adjustment due to aerosols, we consider the sensitivity of scene albedo (A) to $N_d$. $A = A_{ac}Cf_{total} + A_s(1 - Cf_{total})$.

We have:

$$S^* = \frac{dA}{dN_d} = \frac{d(A_{ac}Cf_{total} + A_s(1 - Cf_{total}))}{dN_d} \approx Cf \times S$$
$$+ \left(1 - Cf_{high}\right) \times \frac{dCf}{dN_d} \times (A_{ac} - A_s) \quad (4)$$

where A is the scene albedo, i.e., including both cloudy, $A_{ac}$, and clear, $A_s$, parts; $Cf_{total}$ and $Cf$ are all cloud and low cloud fraction obtained from the MYD08_M3 data; $A_s$ is the surface albedo, derived from the CERES EBAF-TOA data[48]; $1 - Cf_{high}$ is used to take into account of effect of overlap on Cf adjustment. We assume a maximum overlap between high and low clouds. We assume minimum aerosol effects on high clouds. The estimation is done at monthly time scales.

CF and LWP adjustments, $\frac{dCf}{dN_d}$ and $\frac{dlnLWP}{dlnN_d}$, are derived from our previous work based on large number of ship-track samples[21]. The assumption is that clouds with similar properties respond similarly to addition of aerosols and ship-tracks detected under diverse background cloud conditions can be used to effectively derive these adjustments. Our results are based on the responses from observed ship-track sampled under diverse set of environmental conditions, which allows us to derive robust cloud adjustments based on numerous ship-track samples[21]. There are a few assumptions and approximations as noted in our previous study[21] and we reiterate them here. We used $SW_{downwelling}$ at the surface from CERES instead of $SW_{downwelling}$ at the cloud top, which underestimates the total forcing since $SW_{downwelling}$ at the cloud top is larger. The LWP and Cf adjustments can be sensitive to more variables that those considered here. We assume the derived $\frac{dCf}{dN_d}$ and $\frac{dlnLWP}{dlnN_d}$ and their dependence on background variables apply to regions that have less ship-track samples. Also, potential semi-direct effects due to absorbing aerosols from ship-emissions are not explicitly addressed in this study since we do not have enough observations.

The cloud adjustments used here can depend on the background cloud $N_d$, SST, EIS, and background $N_d$ and thus they have spatiotemporal variations due to background changes. The dependence of cloud adjustments can also be parameterized with one or more variables. The $N_d$-only functional form provides a lower bound on the forcing calculation[21] and we explore different 2-variable combinations to provide a range of estimates. We combine the cloud adjustments with the simulated $\Delta N_d$ due to IMO2020 and observations of clouds and other parameters in 2020 to calculate its forcing. We report the mean forcing from all five functional forms of cloud adjustments as well as the standard deviation. We also report the warming effect expected from both the upper and lower bounds in Fig. 3. Due to the $N_d$-dependent nature of cloud adjustments, the same $\Delta N_d$ can result in different magnitude of radiative forcing[21,49].

There is systematic difference between GEOS-modeled and MODIS observed climatology of $N_d$. At each grid point, we calculate the ratio between modeled and observed $N_d$ based on monthly data and scale the modeled $\Delta N_d$ with the ratio before using above equations to calculate the forcing. The global mean values change by 10% between scaled and non-scaled $\Delta N_d$, all coming from the CF adjustment since the LWP adjustment and the Twomey effect depend on $\Delta N_d/N_d$ that does not change with scaling. Regionally, the difference can be as large as 30%, e.g., in the Southeast Atlantic.

The CERES EBAF-TOA data[48] provides monthly and climatological averages of observed top-of-atmosphere and computed cloud radiative effect and absorbed solar radiation. The top-of-atmosphere net fluxes provides constraints to the ocean heat storage. It is used here to calculate the interhemispheric contrast in absorbed solar radiation and energy balance. We note that although the interhemispheric contrast is a residue of two large numbers, e.g., the amount of mean absorbed solar radiation in both hemispheres, the observed variation of the contrast is always small. Therefore, even though we cannot directly attribute the variations in the interhemispheric contrast to IMO 2020, it is reasonable to discuss their temporal evolutions and compare the IMO 2020 impact with the observed changes.

## Transient warming of IMO2020

We consider the simple one-layer energy balance model[27]:

$$C \times \frac{dT}{dt} = F - \lambda \times T \qquad (5)$$

where C is the heat capacity of the well-mixed ocean layer, T is the temperature anomaly from the equilibrium, t is time, F is the forcing, and $\lambda$ is the climate feedback parameter. For an abrupt forcing, the solution is:

$$T = (F/\lambda) \times (1 - e^{-\lambda t / C}) \qquad (6)$$

Using $C = 8.2$ W yr/m2/K and $\lambda = 1.2$ Wm$^{-2}$K$^{-1}$, for a forcing of $F = 0.2$Wm$^{-2}$, we get the temperature change at the new equilibrium is 0.17 K with a time scale of $C/\lambda = 7$ years. The warming rate is $F/C = 0.2/8.2$ K/yr = 0.024 K/yr or 0.24 K/decade. $\lambda$ has uncertainty associated with it and its 1-$\sigma$ is 0.25 Wm$^{-2}$K$^{-1}$. With this, we can estimate equilibrium $T$ to be between 0.14 and 0.21 K. Equation 6 is used to calculate the expected warming trajectory in the 2020 s when combined with a simple long-term upward trend in Fig. 3. The observed global mean temperature is from the National Aeronautics and Space Administration (NASA) Goddard Institute of Space Studies.

## Contributions of background N$_d$, CF, and $\Delta N_d$

We calculate the annual mean of incoming solar radiation for each oceanic grid in the North Atlantic and use this map of seasonally invariant incoming solar radiance to calculate IMO2020 forcing (see section c of Methods). The seasonal cycle of forcing using the seasonally invariant solar radiation is shown in Figure S1, which differs substantially from Fig. 3a, highlighting the impact of seasonal cycle in solar radiation. The peak season for the forcing is now wintertime instead of summertime. This serves as our baseline to test sensitivity of forcing to different variables.

The sensitivity of the forcing to each factor is assessed through the following procedure. We first calculate the seasonal variations of the forcing using observations that contain its seasonal variations. We then calculate a map of annual mean for each variable and use it to calculate the forcing, effectively removing its impact on the seasonal changes. The relative difference between these two calculations can be taken as a measure of how much each variable contributes to the seasonal changes.

## Data availability

MODIS, CERES, and MERRA-2 data are public available at their respective websites: https://ladsweb.modaps.eosdis.nasa.gov/, https://ceres.larc.nasa.gov/, https://gmao.gsfc.nasa.gov/reanalysis/MERRA-2/data_access/. The GOCART simulation data can be found here: https://dataverse.harvard.edu/dataset.xhtml?persistentId=doi:10.7910/DVN/H0ZFK9.

## Code availability

The codes are available here: https://zenodo.org/records/11094677.

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

## Acknowledgements

T.Y. discloses support for the research of this work from NASA TerraAquaNPP program [grant number 80NSSC24K0458], NASA MEaSUReS program [grant number 80NSSC24M0045], NOAA ERB program [grant number NA23OAR4310299] and DOE ASR program [grant number DE-SC0024078].

## Author contributions

T.Y. conceived the idea, designed the experiments, and wrote the draft. H.S. analyzed the data and made plots. H.B. and K.B. ran simulations. T.Y. wrote the manuscript draft and L.O., R.W., M.C., H.Y., D.B, K.M., and S.P. contributed to writing of the manuscript.

## Competing interests

The authors declare no competing interests.
