## [Peer Review File · Communications Earth & Environment]

Web links to the author's journal account have been redacted from the decision letters as indicated to maintain confidentiality.

Decision letter and referee reports: first round

9th Feb 24

Dear Dr Yuan,

Your manuscript titled "Termination Shock of an Inadvertent Geoengineering Experiment Produces Substantial Radiative Warming" has now been seen by 3 reviewers, and we include their comments at the end of this message. They find your work of interest, but some important points are raised. We are interested in the possibility of publishing your study in *Communications Earth & Environment*, but would like to consider your responses to these concerns and assess a revised manuscript before we make a final decision on publication.

We therefore invite you to revise and resubmit your manuscript. Please highlight all changes in the manuscript text file.

In particular, we would like to highlight the need per the reviewer comments for 1) clarification and elaboration of methods; 2) uncertainty bounds on your radiative forcing estimate; and 3) observational comparisons for years outside 2020 as it is impacted by reduced activity due to COVID.

Please use the following link to submit your revised manuscript, point-by-point response to the referees' comments (which should be in a separate document to any cover letter), a tracked-changes version of the manuscript (as a PDF file) and the completed checklist:

[Link Redacted]

We hope to receive your revised paper within six weeks; please let us know if you aren't able to submit it within this time so that we can discuss how best to proceed. If we don't hear from you, and the revision process takes significantly longer, we may close your file. In this event, we will still be happy to reconsider your paper at a later date, as long as nothing similar has been accepted for publication at *Communications Earth & Environment* or published elsewhere in the meantime.

Please do not hesitate to contact us if you have any questions or would like to discuss these revisions further. We look forward to seeing the revised manuscript and thank you for the opportunity to review your work.

Best regards,

Sylvia Sullivan
External Editor
Communications Earth & Environment

Heike Langenberg
Chief Editor
Communications Earth & Environment

EDITORIAL POLICIES AND FORMATTING

Editorial Policy: Policy requirements (Download the link to your computer as a PDF.)

Furthermore, please align your manuscript with our format requirements, which are summarized on the following checklist:

Communications Earth & Environment formatting checklist

and also in our style and formatting guide Communications Earth & Environment formatting guide .

*** DATA: Communications Earth & Environment endorses the principles of the Enabling FAIR data project (<http://www.copdess.org/enabling-fair-data-project/>). We ask authors to make the data that support their conclusions available in permanent, publically accessible data repositories. (Please contact the editor if you are unable to make your data available).

All Communications Earth & Environment manuscripts must include a section titled "Data Availability" at the end of the Methods section or main text (if no Methods). More information on this policy, is available at <http://www.nature.com/authors/policies/data/data-availability-statements-data-citations.pdf>.

If a community resource is unavailable, data can be submitted to generalist repositories such as figshare or Dryad Digital Repository. Please provide a unique identifier for the data (for example a DOI or a permanent URL) in the data availability statement, if possible. If the repository does not provide identifiers, we encourage authors to supply the search terms that will return the data. For data that have been obtained from publically available sources, please provide a URL and the specific data product name in the data availability statement. Data with a DOI should be further cited in the methods reference section.

REVIEWER COMMENTS:

Reviewer #1 (Remarks to the Author):

This is an interesting study of geoengineering termination shock from the sulfur reduction in shipping fuels that mimics naturally occurring geoengineering experiments. The authors use a novel combination of satellite observations, state-of-the-art chemical transport model experiments, machine learning models, and energy balance model to quantify the effect of fuel regulation on cloud

properties, radiative forcing, and warming rate (planetary heat uptake). The warming effect is substantial, to some extent, surprisingly large. The findings have important implications of climate cooling due to anthropogenic aerosols and for marine cloud brightening being a viable geoengineering strategy to counteract greenhouse gas warming. The paper is original and well written. It should be of great interest to others in the community. I only have the following minor comments for clarity:

- 1) In the abstract please make it clear whether those numbers represent global mean or oceanic regional mean. Same for these numbers appearing in the text on Page 5, related to Figure 4.
- 2) "SOM" appears a couple times in the paper but hasn't been defined.
- 3) The purpose of discussion of Tonga volcano eruption is unclear, which actually causes quite bit of confusion to me. Please clarify or remove.
- 4) Page 7 (near the end): warming peak of 1.5 (degree)? By which year?
- 5) The GOCART experiments take MERRA-2 meteorological fields to run in replay mode. I wonder if the inclusion of effects of shipping fuel change on cloud and meteorology in these data has an impact (compensation or double counting) or feedback in the BAU and Covid experiments.
- 6) The sulfur emissions in Table S1 might be better described in units of TgS instead of TgSO₂ and TgDMS.
- 7) Methods b): MAM7 has a primary-carbon mode, which contains hydrophobic black carbon and primary organic carbon. It doesn't make sense to include their mass in the NN algorithm for Nd estimation. Please clarify. Also, why are the secondary organic aerosol and primary organics are combined for this estimation?

Reviewer #2 (Remarks to the Author):

Review of "Termination Shock of an Inadvertent Geoengineering Experiment Produces Substantial Radiative Warming" by Yuan et al

This manuscript combines chemical transport model simulations with observations to try to quantify the impacts of recent shipping regulations that reduce sulfur emissions. The manuscript is generally well written, but I think has some significant methodological problems. I'm not sure it is publishable in its current form.

There are several significant issues. The methodology is not well described or referenced, and is not fully possible to evaluate. Many of the references are not complete: if they are not published yet, this needs to be clearly indicated. Given this, putting it in a format where the methodology is buried at the end of the paper is not really appropriate.

Beyond this, I think the choice of time period (middle of COVID) makes comparison to the observations (which is fundamental to the method) difficult, and potentially misleading. I suggest that observational comparisons need to be done on a different year to make this work.

Furthermore, the translation to simple energy balance models and timescales is also not well treated.

Bottom line: this is quite an interesting piece, but there are some significant issues in trying to make a big claim in a short article. I am not sure this is the right manuscript in this form, and while I support the idea, I think the presentation needs to be rethought and more description of the method (as well as some further experiments to back up the method) presented.

Specific Comments: (page and line number, negative line numbers are up from the bottom of the page).

Page 2, L-14: Suggest removing the reference to figure 1 here. It doesn't really fit and you have not

explained the figure at all.

Page 2,L-8: I think there is spatial structure in some of the analysis (Watson-Parris et al 2022). Also there is structure in earlier reductions (e.,g. Gryspeerdtd 2019).

Watson-Parris, Duncan, Matthew W. Christensen, Angus Laurenson, Daniel Clewley, Edward Gryspeerdtd, and Philip Stier. 2022. "Shipping Regulations Lead to Large Reduction in Cloud Perturbations." *Proceedings of the National Academy of Sciences* 119 (41): e2206885119. <https://doi.org/10.1073/pnas.2206885119>.

Gryspeerdtd, Edward, Tristan W. P. Smith, Eoin O'Keeffe, Matthew W. Christensen, and Fraser W. Goldsworth. 2019. "The Impact of Ship Emission Controls Recorded by Cloud Properties." *Geophysical Research Letters* n/a (n/a). <https://doi.org/10.1029/2019GL084700>.

Page 2, L-3: how is the model run? Nudged? What time period? Are you differencing something. More details please, this is not reproducible. I can find it in the methods (see below), but it makes the work difficult to understand.

Page 3,L+12: What is the mean global Nd? Is 0.5cm-3 significant?

Page 3,L+14: Singular Caribbean, and I don't think the S. China Sea shows much for Nd, there's almost ore effect off New Zealand (which is strange...). Not clear it's only in percent either...Can you show some relationship with ship emissions? Are you sure there is no meteorology effect? I.e, this is run twice with the same meteorology? What happens if you do it for a different year?

Page 3, L-1: Editors decision, but I think there should be some methods explanation here, at least a few sentences.

Having read the methods section, I don't think the methods quite justify this analysis. It's built for one on some work that may be unpublished, and it might need a more complete treatment and discussion.

Page 4, L1: Reference 21 is not complete. Is it published?

Page 4, L1: So if I understand correctly, essentially you use observations of dNd in ship tracts to derive the susceptibilities. Then apply the dNd from the IMO2020 simulations. Are you using one global value? A map? Please put some description here. It's not in the methods either (or not clear in the methods).

Page 4,L14: What is reference 26? A conference paper?

Page 5,L1: This needs more description than just a reference. The methods section has a 1 layer energy balance model. This looks like pieces of it. Please explain the relationship. Does it matter if the warming is on top of an existing trend? Also, it's an instantaneous bump: does it continue forever? To increase the trend?

Page 5, L-4: I am concerned that you are focusing on one region with one year of meteorology here, and yet also doing global extrapolations above. Seems a bit disconnected. Based on the methods, the forcing is coming from an uncertain dNd change. What is the uncertainty on that?

Page 6,L15: where does the 0.25Wm-2 come from if it is 0.05Wm-2/yr?

Please explain the statement "the magnitude of warming is not strong enough to explain 2023" why or why not?

Page 6,L-15: how do you estimate the hemispheres? Just integrate over the forcing from dNd (fig2)?

Page 6,L-9: If I understand 4c to be the interhemispheric contrast, it is LOWER right after IMO2020, and is still lower than the previous few years. I'm not sure your argument makes sense based on those observations?

Page 7,L-12: I think this should start a summary section.

Page 7,L-12: has exceeded. Also, I'm not sure what this whole sentence means.

Page 7,L-6: I don't understand this statement. The inhomogeneity is from the forcing change from ships, it is not 'built in'. Or at least, I don't think you have explained if the background low cloud distribution would make a hemispheric contrast. Please clarify.

Page 8,L1: Is there any evidence that there were any impacts of this hemispheric contrast? Which is also less than the previous years according to 4c.

Page 8,L-12: Awkward jumping in the 2nd to last paragraph from ships to theoretical MCB: suggest that be cleaned up and all the MCB comments put in one place in this summary section.

Page 9,L8: So you are just looking at 2020? I don't think you can use that to make definitive statements. That was massively affected by COVID, so we don't have good emissions estimates. Furthermore, the IMO2020 regulations were phased in over a few months, and then who knows what happened for the next 6 months due to COVID. It can be used as a perturbation, but that's probably not realistic, and not sure you can compare it to observations.

Page 9,L-9: I don't think Google and Apple mobility data works for ships.

Page 10, L20: Reference 46 is not complete. Has it been published? If not, it's a bit premature to use the model this way without a more significant evaluation.

Page 10,L24: Has this version of GOCART been validated? It needs a reference.

Page 10,L-12: Once again: Is reference 21 published or submitted? Cannot be evaluated.

Page 11,L-24: I do not understand this statement about the methodology: it seems a bit circular (no bias from ship tracks because it's based on the response from identified ship tracks). This needs some rewording (and access to reference 21).

Page 11, L-20: here is where I think the method really gets problematic: I don't think you have a good handle on 2020 emissions, and the shipping emissions are evolving that year. Thus I think that the combination of model with observations for 2020 has a lot more uncertainty.

Page 11,L-5: where do these constants come from? No references. Or is it reference 27? Hard to tell from the text. Where does the uncertainty on feedback come from? Also, the section is called "warming rate", but it's not a rate, it's a temperature which I think is equilibrium.

Page 12, L1: Why are you doing this? I'm not clear how this relates to section c. Why are you assessing contributions with the wrong solar forcing when you could use the right forcing? I gather you are trying to sort out seasonal contributions. Why is LWP constant in figure S1? can you please explain that.

Reviewer #3 (Remarks to the Author):

please see the attached .pdf

This manuscript communicates a key finding that the emission regulation implemented on global shipping traffic since 2020 (IMO2020) produced a global radiative forcing of 0.12 W/m^2 through aerosol-cloud interactions and concludes that their results suggest geoengineering approaches such as marine cloud brightening (MCB) may be viable in terms of temporally slowing down the warming rate. This finding is essentially based on modeled changes in global cloud droplet number concentration (N_d) attributed to IMO2020 and the cloud susceptibility (S) assessment by the lead author's previous work using a large number of machine-detected shiptracks and satellite retrieved cloud properties. Besides the key finding, the authors also expand upon the temporal and spatial heterogeneity of this forcing and its significance in the context of the observed recent trend in planetary heat uptake in satellite record.

This work touches on a very intriguing topic (which falls in the scope of the journal), and the authors managed to take advantages of a set of powerful tools, including aerosol chemistry and transport modeling, machine-learning approaches, and satellite observations in a comprehensive and constructive way. Their findings are novel and can potentially provide valuable insights to the climate science community, policy-makers, and the general public. That said, I do find some weaknesses in the current form of the manuscript, especially in the justification of the robustness of these results and the interpretation of their findings.

I find the text well written and easy-to-follow. I recommend that the authors give my comments (listed below) a consideration and justify and/or modify their methodology and interpretation as necessary. Overall, I feel positive about this manuscript and tend to think that it conveys impactful messages worthy of publication, given my concerns are sufficiently addressed.

Major comments: (I numbered the pages, starting from pp. 1 with title, author list, and abstract)

First of all, for this type of high-level, multi-step radiative forcing estimate, I am surprised that the authors did not provide an uncertainty range, given all the assumptions the authors had to make and the fact that susceptibilities are derived from statistical regressions. Without such an uncertainty quantification, this forcing of 0.12 W/m^2 is hard to interpret. Below are some places where I found that I could question the robustness of the results:

- I am convinced that the dCf/dN_d and $dLWP/dN_d$ estimates based on detected-shiptracks represent physical (or causal) responses (pp. 11, second paragraph). However, I see them as causal responses under a specific set of conditions (not all conditions) that are favorable for detectable shiptracks. I believe causal aerosol effects also exist in undetected tracks which need to be accounted as well when generalizing. Therefore, I think the caveat here is that while dN_d represent all conditions, susceptibilities used in the calculations are only for conditions favorable for detectable shiptracks (e.g., strong LTS). I suggest adding related discussion to the main text.
- Related to the above point, the assumption to approximate susceptibility as a 1D function of N_d is a bit concerning to me, as we know cloud susceptibility is highly dependent on the governing large-scale meteorological conditions (many references in existing literature, e.g., Gryspeerd et al., Toll et al., Possner et al., ...). In fact, in ref. 21 where these susceptibilities are first derived by the lead author, the narrow spread in dCf/dN_d and $dLWP/dN_d$ as a function of N_d (Fig. 4&5 therein) suggests to me that detected shiptracks only occur in certain

combinations of large-scale conditions (or there is a strong covariation between N_d and large-scale conditions when shiptracks are detectable). I feel the authors should better justify or mention the limitation of this assumption in the text.

- The study by Diamond (2023) has been brought up a few times in the manuscript where the authors state that their estimate of the total aerosol effect of $\sim 0.5 \text{ W/m}^2$ over the SE Atlantic shipping corridor is in “*excellent*” agreement with Diamond (2023). However, Diamond (2023) estimated the IRF_{ACI} (equivalent to the Twomey component in this study) to be $\sim 0.5 \text{ W/m}^2$, and did not address cloud adjustments. This suggests a discrepancy between the two studies, instead of an agreement.
- The scaling of dN_d (*Methods*, part (c)), to me, seems unjustified and unnecessary. It’s not clear to me how and why the difference between 2020 MODIS- N_d and GEOS- N_d is related to GEOS-“d” N_d such that one can use the ratio between absolute values to scale a difference.
 - o I believe that the transport model is used for the purpose of getting N_d changes, rather than reproducing MODIS- N_d (GEOS- N_d maybe biased, but GEOS- dN_d could be realistic; if there’s bias in the model, shouldn’t the bias be cancelled out by taking the difference to get dN_d ?);
 - o Moreover, shouldn’t the 2020 MODIS- N_d be closer to the effect of COVID+IMO2020, whereas you modeled GEOS- N_d is only accounting for IMO2020?
 - o In my view, if one wants to get an observational constraint on dN_d , one could get MODIS derived dN_d by taking the difference between 2020-MODIS- N_d and 2015-2019 MODIS climatology for example; (perhaps a simple comparison here between the two dN_d could provide some insights)
 - o All that being said, if one quantifies $dC_f/d\ln N_d$ as susceptibility, one wouldn’t need to worry about this in the first place, right?

Some parts of the *Methods* are unclear to me,

- I feel that the introduction to the setup of the GOCART runs is hard to follow for someone who’s not familiar with an aerosol transport model, in particular the first paragraph on pp. 9. For instance, what do you mean by “... *atmospheric state from a priori analysis* ...”, what is that “*priori analysis*”, what is a “*model incremental analysis update*”, and what do you mean by “*forecast model*” (same GOCART model in forecast mode)? More context and plainer language here would help a lot, I think.
- What’s the grid-spacing and time step for the transport model? What temporal resolution is used in the forcing calculations?
- dN_d is shown in Figure 1 and used throughout the study to calculate the radiative forcing, but it’s unclear how did the authors actually calculated dN_d from their experiments, i.e., which 2 scenarios did the authors actually use to take the difference to get dN_d ?
- There is no mention of the COVID experiments and their purpose at all throughout the main text.
- Eqn. 4 is not clear/straightforward to me, what’s your assumption on the relationship among C_{tot} , C_{high} , and C_f , and how does $(1-C_{\text{high}})$ account for the effect of overlap (of what)?

Another concern is that the authors tend to establish the significance of the IMO2020-forcing based on comparisons against averaged/net trends in warming or heat uptake, which are results of forcings from multiple agents that are often opposite in sign. I understand the rationale when all

forcings have the same sign. However, without knowing the contributions from other forcings at play, such a way to establish significance could be misleading. (A simple example: a net forcing of 10 W/m² is consist of forcing A of 30 W/m², forcing B of -30 W/m², and forcing C of 10 W/m², clearly, it's unfair to say forcing C contribute 100% to the net forcing).

- Abstract, “*The warming effect contributes 50% to the measured increase in planetary heat uptake since 2020.*” This attribution is clearly not supported by the analysis shown in this study, and even the authors later mention this point on pp. 6 second paragraph, “*the comparison does not prove causality but provides a context...*”
- Regarding pp. 6 second paragraph, I question the necessity of this discussion, which is based on speculations rather than causal attributions, and one can't learn much from these statements such as “*...represents around 40% of increase in NH...*” and “*...suggest as much as 60% increase in interhemispheric...*” that are not factual attributions.
- Another similar example is on pp. 5, line 5, “*...40% higher than the average warming rate since 1880...*”, to me, it would be more relevant if one compares against the historical anthropogenic aerosol (AA) cooling rate rather than the average warming rate (a lesser magnitude due to AA cooling cancelling GHG warming).

More generally, uncertainty can source from the nature of nonlinearity in aerosol-cloud interactions (e.g., Gryspeerdt et al. 2023, Jia and Quaas 2023). For example, the authors use S derived from $+N_d$ perturbation (shiptrack) experiments (statistical regressions assuming linear response) to predict cloud responses to $-N_d$, which can be subject to the influence of nonlinearity. Moreover, the main conclusion or implication of the study that MCB is a viable geoengineering scheme is mainly based on the assumption that the strong radiative forcing due to $-N_d$ (IMO2020) can be mirrored to a $+N_d$ scenario (e.g., MCB), again, relying on the assumption of a linear system. We know, however, aerosol indirect effects tend to saturate at high N_d (where already bright clouds cannot be brightened anymore). Given that we are still in a polluted (high- N_d) world, the actual cooling generated by $+N_d$ might be much smaller than the warming from cleaning up the same amount N_d . For these reasons, I feel the authors need to justify their main conclusion in the context of nonlinearity.

Some minor notes:

- pp. 2, paragraph 3, line 2, I wouldn't call shipping emission as “*long-running inadvertent MCB experiment,*” as the composition of the ship exhaust can be quite different from what the community has proposed to use in the context of MCB (i.e., sea salt particles).
- pp. 2, paragraph 3, line 6, “*... by dimming clouds*” I believe you meant dimming low clouds, not all clouds, I would be more precise here.
- Figure 1, I recommend adding some details to the caption, e.g., this is dN_d averaged over the year of 2020 (right?). On c), is $d\ln(N_d)$ actually shown, the color bar confuses me a little, is the maximum change of N_d only $\sim 3/cc$?
- pp. 3, third to the last line, how is ref. 16 relevant to this sentence?
- pp. 4, third to the last line, what does SOM represent?
- Figure 2, caption, “*... forcing from cloud dimming,*” not all components are dimming the clouds, I see a significant portion of LWP adjustments actually contributes to brightening. Could you also indicate in *Methods* how you quantify the Twomey and LWP contributions separately based on Eqn. 3.

- pp. 5, 4 lines above the figure, “*likely*” or “*possibly*”? Are there observational evidence of low-cloud albedo decreases in the North Atlantic since 2020, e.g., in satellite record (CERES)?
- Figure 3 & 4, font is inconsistent with Figure 1 & 2 and looks a bit small. Figure 4 caption, “*Details in Methods section*” I did not see a relevant introduction of the CERES data in *Methods*.
- Pp. 6, line 1-2, I think reference(s) suggesting that interhemispheric contrast in forcing leads to substantial changes in monsoons and precipitation patterns is needed here.
- Pp. 6, forth to the last line, “... *in the contrast instead of ...*” please check the wording.
- Pp. 6, paragraph 2, on dN_a uncertainty, is it cost-efficient to run a GOCART ensemble to estimate this uncertainty? just curious...
- I see Supporting Materials attached to the end, but they are not referenced at all in the main manuscript. Is this on purpose or something is missing?

References

Gryspeerdt, E., Povey, A. C., Grainger, R. G., Hasekamp, O., Hsu, N. C., Mulcahy, J. P., Sayer, A. M., and Sorooshian, A.: Uncertainty in aerosol–cloud radiative forcing is driven by clean conditions, *Atmos. Chem. Phys.*, 23, 4115–4122, <https://doi.org/10.5194/acp-23-4115-2023>, 2023.

Jia, H., Quaas, J. Nonlinearity of the cloud response postpones climate penalty of mitigating air pollution in polluted regions. *Nat. Clim. Chang.* 13, 943–950 (2023). <https://doi.org/10.1038/s41558-023-01775-5>

Author Responses: first round

We appreciate all comments and suggestions by the reviewers as they helped us improve our manuscript. We performed additional calculations to quantify both the uncertainty and range of our estimate using the other functional forms of cloud adjustments and uncertainty range of each adjustment. The estimated mean and standard deviation of forcing calculated based on alternative adjustment forms are $+0.2 \pm 0.11 \text{ Wm}^{-2}$. Based on these new calculations, we updated our abstract and conclusion as well as parts of the main text. We added a new figure to address the expected warming effect in the 2020s from the IMO 2020. We now plot an estimated mean warming trajectory in the 2020s with lower and upper bounds corresponding to lower and upper bounds of radiative forcing induced by IMO 2020. We also calculated the forcing using 2021 conditions to test the sensitivity of results to interannual variations as suggested by reviewers.

In the following, we provide point-by-point responses.

Reviewer #1 (Remarks to the Author):

This is an interesting study of geoengineering termination shock from the sulfur reduction in shipping fuels that mimics naturally occurring geoengineering experiments. The authors use a novel combination of satellite observations, state-of-the-art chemical transport model experiments, machine learning models, and energy balance model to quantify the effect of fuel regulation on cloud properties, radiative forcing, and warming rate (planetary heat uptake). The warming effect is substantial, to some extent, surprisingly large. The findings have important implications of climate cooling due to anthropogenic aerosols and for marine cloud brightening being a viable geoengineering strategy to counteract greenhouse gas warming. The paper is original and well written. It should be of great interest to others in the community. I only have the following minor comments for clarity:

1) In the abstract please make it clear whether those numbers represent global mean or oceanic regional mean. Same for these numbers appearing in the text on Page 5, related to Figure 4.

Response: Thank you! This is an important point. In the abstract, we did state that the forcing represents a global ocean mean value. We now explicitly state this in the text as well when we discuss the forcing number for the first time.

2) "SOM" appears a couple times in the paper but hasn't been defined.

Response: Now defined.

3) The purpose of discussion of Tonga volcano eruption is unclear, which actually causes quite bit of confusion to me. Please clarify or remove.

Response: Removed.

4) Page 7 (near the end): warming peak of 1.5 (degree)? By which year?

Response: 1.5K is the more aspirational goal. It does not have a time limit but the corresponding emission controls have time limits due to this goal.

5) The GOCART experiments take MERRA-2 meteorological fields to run in replay mode. I wonder if the inclusion of effects of shipping fuel change on cloud and meteorology in these data

has an impact (compensation or double counting) or feedback in the BAU and Covid experiments.

Response: The model is indeed run in replay mode. We essentially drive the model with meteorology from reanalysis and no feedback from cloud changes affect meteorology. Clearer language is now used to describe the replay mode.

6) The sulfur emissions in Table S1 might be better described in units of TgS instead of TgSO₂ and TgDMS.

Response: Good idea. Changed accordingly.

7) Methods b): MAM7 has a primary-carbon mode, which contains hydrophobic black carbon and primary organic carbon. It doesn't make sense to include their mass in the NN algorithm for Nd estimation. Please clarify. Also, why are the secondary organic aerosol and primary organics are combined for this estimation?

Response: It also contains sulfate, so there is potential for Na₂SO₄ formation. We don't make a priori assumptions about what mode can nucleate droplets. Instead, it is dictated by the modal hygroscopicity (κ) calculated as the volume-weighted average of κ for each species in each mode. Black carbon is assumed to have $\kappa=0.001$, hence by itself would not nucleate droplets. The algorithm was intended to use MERRA-2 aerosol mass as input. MERRA-2 does not distinguish between primary and secondary organic aerosol. MAMnet however is able to discriminate between primary and secondary aerosol and assign mass to the proper mode.

Reviewer #2 (Remarks to the Author):

Review of "Termination Shock of an Inadvertent Geoengineering Experiment Produces Substantial Radiative Warming" by Yuan et al

This manuscript combines chemical transport model simulations with observations to try to quantify the impacts of recent shipping regulations that reduce sulfur emissions. The manuscript is generally well written, but I think has some significant methodological problems. I'm not sure it is publishable in its current form.

There are several significant issues. The methodology is not well described or referenced, and is not fully possible to evaluate. Many of the references are not complete: if they are not published yet, this needs to be clearly indicated. Given this, putting it in a format where the methodology is buried at the end of the paper is not really appropriate.

Response: We have updated the references the best we could. The details of our method are described in Yuan et al. which was accepted but not published at the time this manuscript was

submitted, but is now published. Our method is also similar to Toll et al. (2019) except for the additional inclusion of of the CF adjustment effect.

Beyond this, I think the choice of time period (middle of COVID) makes comparison to the observations (which is fundamental to the method) difficult, and potentially misleading. I suggest that observational comparisons need to be done on a different year to make this work.

Response: We would like to clarify that our results do not attempt to compare with observed low cloud dimming forcing for which we have no observations. We rather use a hybrid approach where modeling is employed to estimate dN_a due to IMO 2020 and cloud adjustments are based on observational results from Yuan et al. (2023). We performed simulations with emission baselines (i.e. no IMO 2020 impact) for both COVID and no-COVID-impact conditions. The results differ by 10%, which is now presented in the manuscript.

Furthermore, the translation to simple energy balance models and timescales is also not well treated.

Response: We applied the estimated forcing to the energy balance model and calculated the resulting warming effect, which is how a model of this type is typically used. We are not sure what the reviewer's objections are without additional details.

Bottom line: this is quite an interesting piece, but there are some significant issues in trying to make a big claim in a short article. I am not sure this is the right manuscript in this form, and while I support the idea, I think the presentation needs to be rethought and more description of the method (as well as some further experiments to back up the method) presented.

Response: Our method is based on a published study (Yuan et al., 2023), which is in turn based on well-established previous studies. We are addressing the reviewer's specific comments below.

Specific Comments: (page and line number, negative line numbers are up from the bottom of the page).

Page 2, L-14: Suggest removing the reference to figure 1 here. It doesn't really fit and you have not explained the figure at all.

Response: There may be some format mismatch between what reviewer sees and the online PDF because we do not see reference to figure 1 at page 2 line 14. If the reviewer is referring to the reference after '... marine cloud dimming (...)', we modified it slightly to make clear what we mean here.

Page 2,L-8: I think there is spatial structure in some of the analysis (Watson-Parris et al 2022). Also there is structure in earlier reductions (e.,g. Gryspeerdt 2019).

Response: Again, there may be a mismatch in the format of the manuscript. On page2 Line 8, there is no reference for a spatial structure. We are not sure what 'spatial structure in some of the

analysis' refers to. If the reviewer refers to the ocean near California, this is indeed an important region because of high shipping traffic.

Watson-Parris, Duncan, Matthew W. Christensen, Angus Laurenson, Daniel Clewley, Edward Gryspeerdt, and Philip Stier. 2022. "Shipping Regulations Lead to Large Reduction in Cloud Perturbations." *Proceedings of the National Academy of Sciences* 119 (41): e2206885119. <https://doi.org/10.1073/pnas.2206885119>.

Gryspeerdt, Edward, Tristan W. P. Smith, Eoin O'Keeffe, Matthew W. Christensen, and Fraser W. Goldsworth. 2019. "The Impact of Ship Emission Controls Recorded by Cloud Properties." *Geophysical Research Letters* n/a (n/a). <https://doi.org/10.1029/2019GL084700>.

Page 2, L-3: how is the model run? Nudged? What time period? Are you differencing something. More details please, this is not reproducible. I can find it in the methods (see below), but it makes the work difficult to understand.

Response: In the method section (a), we stated that the model is run in replay mode and described how it is run. We added a description on how the difference is produced at the end of this subsection, i.e., Method section (a).

Page 3, L+12: What is the mean global Nd? Is 0.5cm⁻³ significant?

Response: To give a context, the modeled global mean N_d is 28cm^{-3} . So the mean change is about 2%, but the change is spatially unevenly distributed.

Page 3, L+14: Singular Caribbean, and I don't think the S. China Sea shows much for Nd, there's almost ore effect off New Zealand (which is strange...). Not clear it's only in percent either....Can you show some relationship with ship emissions? Are you sure there is no meteorology effect? I.e, this is run twice with the same meteorology? What happens if you do it for a different year?

Response: If the reviewer refers to relative change, it is indeed large off the coast of New Zealand. This is a combination of cleaner background in that region and contribution from shipping emission around Australia and New Zealand. In terms of absolute dN_d , values in the South China Sea are larger. The with and without IMO2020 runs are made with the same meteorology since they are conducted in replay mode.

Page 3, L-1: Editors decision, but I think there should be some methods explanation here, at least a few sentences.

Having read the methods section, I don't think the methods quite justify this analysis. It's built for one on some work that may be unpublished, and it might need a more complete treatment and discussion.

Response: The method is described in Yuan et al. (2023) which has now been published. It builds on well-established principles that are described in this manuscript. For example, it is similar to Toll et al. (2019) but includes the effect of CF adjustments.

Page 4, L1: Reference 21 is not complete. Is it published?

Response: We modified the reference 21. At submitting time, this paper was accepted but not published. Now it is.

Page 4, L1: So if I understand correctly, essentially you use observations of dN_d in ship tracts to derive the susceptibilities. Then apply the dN_d from the IMO2020 simulations. Are you using one global value? A map? Please put some description here. It's not in the methods either (or not clear in the methods).

Response: We apply the cloud adjustment relationships from ship-track analyses (Yuan et al., 2023). Yes, we then use simulated dN_d induced by changes in ship emissions due to IMO2020, and other observations and reanalysis data to estimate the forcing. The adjustments are state-dependent. Here we use its dependence on only background cloud N_d . This provides a lower bound on the forcing magnitude since dependence on other factors, documented in reference 21, can make it larger. We have now also added results from additional calculations based on cloud adjustment dependences on co-variation of two variables such as N_d and SST, N_d and estimated inversion strength, and N_d and cloud fraction. We added a description of these calculations in the method subsection C.

Page 4, L14: What is reference 26? A conference paper?

Response: Yes.

Page 5, L1: This needs more description than just a reference. The methods section has a 1 layer energy balance model. This looks like pieces of it. Please explain the relationship. Does it matter if the warming is on top of an existing trend? Also, it's an instantaneous bump: does it continue forever? To increase the trend?

Response: Yes, we effectively used a one-layer model by ignoring the heat exchange between the mixed layer and deep ocean, which is now stated in the text. This model considers the impact of the additional warming which has an equilibrium response as well as a timescale associated with it. In method subsection d, we provide more details.

Page 5, L-4: I am concerned that you are focusing on one region with one year of meteorology here, and yet also doing global extrapolations above. Seems a bit disconnected. Based on the methods, the forcing is coming from an uncertain dN_d change. What is the uncertainty on that?

Response: We do that indeed. Using the North Atlantic as an example should not affect generality. The analysis is to show the impact of different factors that contribute to the forcing.

The conclusion is very likely applicable to other regions because of equation 4. Here we use the North Atlantic as an example to quantify the contribution of several factors. We show that dN_d is the most important factor in determining the forcing variation. We estimate the impact of the potential range/uncertainty of dN_d by using two baselines: business as usual and with COVID reduction as stated in the method section. We have now also added uncertainty estimate of the forcing due to uncertainty in the cloud adjustments based on Yuan et al. (2023).

Page 6,L15: where does the 0.25Wm^{-2} come from if it is $0.05\text{Wm}^{-2}/\text{yr}$?

Response: 0.05Wm^{-2} refers to the long-term mean. 0.25Wm^{-2} refers to the total change since 2020. We stated this in the text.

Please explain the statement “the magnitude of warming is not strong enough to explain 2023” why or why not?

Response: Given the new calculations of forcing and expected warming, we have removed the sentence and added more discussion based on the new results. Now, we believe that IMO 2020 impact is a substantial contributor to the anomalously warm year in 2023. We also expect that 2020's will remain anomalously warm because of the radiative forcing.

Page 6,L-15: how do you estimate the hemispheres? Just integrate over the forcing from dN_d (fig2)?

Response: Yes, we use an area-weighted mean of calculated forcing in both hemispheres and calculate the contrast.

Page 6,L-9: If I understand 4c to be the interhemispheric contrast, it is LOWER right after IMO2020, and is still lower than the previous few years. I'm not sure your argument makes sense based on those observations?

Response: The reviewer is correct that during part of 2020 the contrast was low and still trending downward. This is likely because of the strong downward trend that begun several years before that and reflects the inertia of the system. It then had a swift reverse during and after IMO 2020, which would be consistent with the strong warming effect from IMO 2020.

Page 7,L-12: I think this should start a summary section.

Response: We are not sure about the recommendation here, but we did rearrange some text.

Page 7,L-12: has exceeded. Also, I'm not sure what this whole sentence means.

Response: Changed to 'exceeded'. We also modified the text to make it clearer.

Page 7,L-6: I don't understand this statement. The inhomogeneity is from the forcing change

from ships, it is not 'built in'. Or at least, I don't think you have explained if the background low cloud distribution would make a hemispheric contrast. Please clarify.

Response: We agree with the reviewer that 'built in' is inappropriate in this context. We changed it to 'Important part of...'. Here we are only referring to the inhomogeneity of cloud fields, not commenting on the hemispheric contrast.

Page 8,L1: Is there any evidence that there were any impacts of this hemispheric contrast? Which is also less than the previous years according to 4c.

Response: We do not know the answer to this question. The point we are trying to make here is that a geoengineering experiment needs to consider this contrast because it can have important consequences based on previous studies. What matters is the perturbation an experiment could cause.

Page 8,L-12: Awkward jumping in the 2nd to last paragraph from ships to theoretical MCB: suggest that be cleaned up and all the MCB comments put in one place in this summary section.

Response: In the second to last paragraph, we discuss uncertainties that are involved in the estimate. That is how this paragraph was organized. We think that it is appropriate, and if the reviewer agrees, we would like to keep it as is.

Page 9,L8: So you are just looking at 2020? I don't think you can use that to make definitive statements. That was massively affected by COVID, so we don't have good emissions estimates. Furthermore, the IMO2020 regulations were phased in over a few months, and then who knows what happened for the next 6 months due to COVID. It can be used as a perturbation, but that's probably not realistic, and not sure you can compare it to observations.

Response: In this manuscript, we ran experiments based on 2020 meteorology. We considered the impact of COVID by carrying out different baselines: BAU and COVID. We may indeed have an imperfect COVID emission impact yet as the reviewer points out. We are running our experiments based on current best estimates. We are not comparing simulations to observations. The GOCART simulation is performed to estimate dN_d which is otherwise hard to obtain observationally. As suggested by the reviewer, we also calculate the forcing using 2021 data and the results show small sensitivity.

Page 9,L-9: I don't think Google and Apple mobility data works for ships.

Response: Agreed. We want to note that they are only used for land-based emissions to approximate the impact of COVID.

Page 10, L20: Reference 46 is not complete. Has it been published? If not, it's a bit premature to use the model this way without a more significant evaluation.

Response: Yes, it is now published here: <https://journals.ametsoc.org/view/journals/aies/3/1/AIES-D-23-0025.1.xml>

Page 10,L24: Has this version of GOCART been validated? It needs a reference.

Response: We are now providing two references. The GOCART model is well-established and accepted by the community, and has been shown to perform well for purposes similar to the one it is used here for.

Page 10,L-12: Once again: Is reference 21 published or submitted? Cannot be evaluated.

Response: It is published and open-access:
<https://www.science.org/doi/10.1126/sciadv.adh7716> .

Page 11,L-24: I do not understand this statement about the methodology: it seems a bit circular (no bias from ship tracks because it's based on the response from identified ship tracks). This needs some rewording (and access to reference 21).

Response: Indeed, the sentence is a bit confusing. We have rearranged it to make the point clearer.

Page 11, L-20: here is where I think the method really gets problematic: I don't think you have a good handle on 2020 emissions, and the shipping emissions are evolving that year. Thus I think that the combination of model with observations for 2020 has a lot more uncertainty.

Response: We are using the best emission estimate available, which is a standard practice. We thus do not believe that this problematic. We do agree that there are uncertainties to the estimate as we discuss in the text and also in reference 21. Such potential uncertainties are not prohibitive for carrying out the work herein in our opinion.

Page 11,L-5: where do these constants come from? No references. Or is it reference 27? Hard to tell from the text. Where does the uncertainty on feedback come from? Also, the section is called "warming rate", but it's not a rate, it's a temperature which I think is equilibrium.

Response: They come from reference 27. We moved the reference number to make it clear. We removed rate from the subsection title.

Page 12, L1: Why are you doing this? I'm not clear how this relates to section c. Why are you assessing contributions with the wrong solar forcing when you could use the right forcing? I gather you are trying to sort out seasonal contributions. Why is LWP constant in figure S1? can you please explain that.

Response: We are indeed highlighting the importance of considering seasonally varying solar forcing. One could just use annual mean solar constant, which would introduce error compared to seasonally varying solar constants. The LWP is not constant; its variation is not as notable because it is smaller than other components.

Reviewer #3 (Remarks to the Author):

This manuscript communicates a key finding that the emission regulation implemented on global shipping traffic since 2020 (IMO2020) produced a global radiative forcing of 0.12 W/m^2 through aerosol-cloud interactions and concludes that their results suggest geoengineering approaches such as marine cloud brightening (MCB) may be viable in terms of temporally slowing down the warming rate. This finding is essentially based on modeled changes in global cloud droplet number concentration (N_d) attributed to IMO2020 and the cloud susceptibility (S) assessment by the lead author's previous work using a large number of machine-detected shiptracks and satellite retrieved cloud properties. Besides the key finding, the authors also expand upon the temporal and spatial heterogeneity of this forcing and its significance in the context of the observed recent trend in planetary heat uptake in satellite record.

This work touches on a very intriguing topic (which falls in the scope of the journal), and the authors managed to take advantages of a set of powerful tools, including aerosol chemistry and transport modeling, machine-learning approaches, and satellite observations in a comprehensive and constructive way. Their findings are novel and can potentially provide valuable insights to the climate science community, policy-makers, and the general public. That said, I do find some weaknesses in the current form of the manuscript, especially in the justification of the robustness of these results and the interpretation of their findings.

I find the text well written and easy-to-follow. I recommend that the authors give my comments (listed below) a consideration and justify and/or modify their methodology and interpretation as necessary. Overall, I feel positive about this manuscript and tend to think that it conveys impactful messages worthy of publication, given my concerns are sufficiently addressed.

Major comments: *(I numbered the pages, starting from pp. 1 with title, author list, and abstract)*

First of all, for this type of high-level, multi-step radiative forcing estimate, I am surprised that the authors did not provide an uncertainty range, given all the assumptions the authors had to make and the fact that susceptibilities are derived from statistical regressions. Without such an uncertainty quantification, this forcing of 0.12 W/m^2 is hard to interpret. Below are some places where I found that I could question the robustness of the results:

- I am convinced that the dC_f/dN_d and $dLWP/dN_d$ estimates based on detected-shiptracks represent physical (or causal) responses (pp. 11, second paragraph). However, I see them as causal responses under a specific set of conditions (not all conditions) that are favorable for detectable shiptracks. I believe causal aerosol effects also exist in undetected tracks which need to be accounted as well when generalizing. Therefore, I think the caveat here is that while dN_d represent all conditions,*

susceptibilities used in the calculations are only for conditions favorable for detectable shiptracks (e.g., strong LTS). I suggest adding related discussion to the main text.

Response:

This issue was raised in the review process of Yuan et al. (2023) and we addressed it there. Essentially, we show that detected ship-track samples cover broad range of conditions (including stability conditions captured by measures such as estimated inversion strength and SST etc.) and the assumption is that under similar conditions, aerosols have similar impact, which is a reasonable assumption in our opinion and stated in the text. We refer to that paper in the current manuscript for more detailed discussions about this topic.

- - *Related to the above point, the assumption to approximate susceptibility as a 1D function of N_d is a bit concerning to me, as we know cloud susceptibility is highly dependent on the governing large-scale meteorological conditions (many references in existing literature, e.g., Gryspeerdt et al., Toll et al., Possner et al., ...). In fact, in ref. 21 where these susceptibilities are first derived by the lead author, the narrow spread in dC_f/dN_d and $dLWP/dN_d$ as a function of N_d (Fig. 4&5 therein) suggests to me that detected shiptracks only occur in certain combinations of large-scale conditions (or there is a strong covariation between N_d and large-scale conditions when shiptracks are detectable). I feel the authors should better justify or mention the limitation of this assumption in the text.*

Response:

This is indeed an assumption made in our calculation. But it also represents a lower bound in terms of total forcing. We added this point in our text and performed additional calculations by considering the dependence of low cloud adjustments on other variables such as estimated inversion strength, sea surface temperature, and background low cloud fraction. We now report the mean of these ensemble calculations. It is worth noting that a similar assumption was made elsewhere as it captures one of the most important controlling variables (e.g., Toll et al. (2019)). In Yuan et al. (2023) we discussed this point in more detail.

- - *The study by Diamond (2023) has been brought up a few times in the manuscript where the authors state that their estimate of the total aerosol effect of $\sim 0.5 \text{ W/m}^2$ over the SE Atlantic shipping corridor is in “excellent” agreement with Diamond (2023). However, Diamond (2023) estimated the IRFACI (equivalent to the Twomey component in this study) to be $\sim 0.5 \text{ W/m}^2$, and did not address cloud adjustments. This suggests a discrepancy between the two studies, instead of an agreement.*

Response:

The reviewer is correct that Diamond (2023) focused on the Twomey effect because he assumed it is dominant in this region. The Twomey effect also remains the dominant

component in the main shipping lane of the Southeast Atlantic in our calculations, so the agreement with Diamond (2023) is not invalidated with our results. We have modified the text to clarify this point.

- - *The scaling of dN_d (Methods, part (c)), to me, seems unjustified and unnecessary. It's not clear to me how and why the difference between 2020 MODIS- N_d and GEOS- N_d is related to GEOS- " d " N_d such that one can use the ratio between absolute values to scale a difference. I believe that the transport model is used for the purpose of getting N_d changes, rather than reproducing MODIS- N_d (GEOS- N_d maybe biased, but GEOS- dN_d could be realistic; if there's bias in the model, shouldn't the bias be cancelled out by taking the difference to get dN_d ?); Moreover, shouldn't the 2020 MODIS- N_d be closer to the effect of COVID+IMO2020, whereas you modeled GEOS- N_d is only accounting for IMO2020?*

Response:

One reason for scaling is that modeled N_d exhibits systematic biases compared to MODIS. This bias is in the climatology, which similarly exists in many global models, and does not pertain to only the 2020 data. Second, the cloud adjustments are based on MODIS observations and they depend on MODIS-based N_d estimates. One way to keep them consistent is thus to scale the modeled N_d . The LWP adjustment is insensitive to the absolute change, but CF adjustment is not. Discussions about this were included in the manuscript. The impact of using scaled vs non-scaled data is about 10%.

o In my view, if one wants to get an observational constraint on dN_d , one could get MODIS derived dN_d by taking the difference between 2020-MODIS- N_d and 2015-2019 MODIS climatology for example; (perhaps a simple comparison here between the two dN_d could provide some insights)

Response: We feel that if were to do that too many sources of errors and uncertainties would be embedded to attribute the difference to IMO 2020. We thus prefer our modeling approach here.

o All that being said, if one quantifies $dC_f/d\ln N_d$ as susceptibility, one wouldn't need to worry about this in the first place, right?

Response: The reviewer is correct about if $\ln N_d$ were to be used, i.e., relative changes. However, physically, the CF adjustment should not scale with relative changes in N_d because this adjustment has a threshold behavior that depends on absolute values. That is why we opt to use the absolute change in reference 21 to calculate the adjustments.

Some parts of the Methods are unclear to me,

- - *I feel that the introduction to the setup of the GOCART runs is hard to follow for someone who's not familiar with an aerosol transport model, in particular the first paragraph on pp. 9. For instance, what do you mean by "... atmospheric state from a*

priori analysis ...”, what is that “priori analysis”, what is a “model incremental analysis update”, and what do you mean by “forecast model” (same GOCART model in forecast mode)? More context and plainer language here would help a lot, I think.

- - *What’s the grid-spacing and time step for the transport model? What temporal resolution is used in the forcing calculations?*

Response: Thanks for the suggestion. We rewrote this introduction using more plain language and added the resolution of the model.

- - *dNd is shown in Figure 1 and used throughout the study to calculate the radiative forcing, but it’s unclear how did the authors actually calculated dNd from their experiments, i.e., which 2 scenarios did the authors actually use to take the difference to get dNd?*

Response: We take the difference between N_d 's with and without the reduction of sulfur emission from IMO 2020.

- - *There is no mention of the COVID experiments and their purpose at all throughout the main text.*

Response: The reviewer is right. We mentioned the difference if considering COVID impact in the method section and it is small. We did not discuss it in the main text given the small perturbation.

- - *Eqn. 4 is not clear/straightforward to me, what’s your assumption on the relationship among C_{tot} , C_{high} , and C_f , and how does $(1-C_{high})$ account for the effect of overlap (of what)?*

Response: Yes, this accounts for the effect of maximum overlap we assumed in reference 21, which details all the assumptions and derivations.

Another concern is that the authors tend to establish the significance of the IMO2020-forcing based on comparisons against averaged/net trends in warming or heat uptake, which are results of forcings from multiple agents that are often opposite in sign. I understand the rationale when all forcings have the same sign. However, without knowing the contributions from other forcings at play, such a way to establish significance could be misleading. (A simple example: a net forcing of 10 W/m^2 is consist of forcing A of 30 W/m^2 , forcing B of -30 W/m^2 , and forcing C of 10 W/m^2 , clearly, it’s unfair to say forcing C contribute 100% to the net forcing).

Response: We understand the reviewer’s concern. However, the SH and NH are very balanced in terms of energy fluxes and there are many studies that try to address the remarkably small contrast in the energy balance of the hemispheres. Given these facts, we always expect relatively small changes in this contrast and thus the IMO 2020 impact is an

important factor. However, we took the point into consideration and changed the language to reflect this concern at the end of method section c.

- - *Abstract, “The warming effect contributes 50% to the measured increase in planetary heat uptake since 2020.” This attribution is clearly not supported by the analysis shown in this study, and even the authors later mention this point on pp. 6 second paragraph, “the comparison does not prove causality but provides a context...”*

Response: We change ‘contributes’ to ‘represents’ to avoid the direct attribution implication. Please see the changes we made in response to the previous comment. Here our main point is that compared to the observed changes in the contrast, the impact from IMO 2020 is not negligible.

- - *Regarding pp. 6 second paragraph, I question the necessity of this discussion, which is based on speculations rather than causal attributions, and one can’t learn much from these statements such as “...represents around 40% of increase in NH...” and “...suggest as much as 60% increase in interhemispheric...” that are not factual attributions.*

Response: We agree that this is not causal, which is clearly stated in the text. But we disagree that this is pure speculation. We are comparing the estimates from our calculations and observations. Although we cannot directly connect the observations to the calculations, it is reasonable to compare them and discuss their implications.

- - *Another similar example is on pp. 5, line 5, “...40% higher than the average warming rate since 1880...”, to me, it would be more relevant if one compares against the historical anthropogenic aerosol (AA) cooling rate rather than the average warming rate (a lesser magnitude due to AA cooling cancelling GHG warming).*

Response: We are not sure why this is not an appropriate comparison. We are placing the additional warming introduced by the IMO 2020 into the long-term perspective.

More generally, uncertainty can source from the nature of nonlinearity in aerosol-cloud interactions (e.g., Gryspeerd et al. 2023, Jia and Quaas 2023). For example, the authors use S derived from +Nd perturbation (shiptrack) experiments (statistical regressions assuming linear response) to predict cloud responses to -Nd, which can be subject to the influence of nonlinearity. Moreover, the main conclusion or implication of the study that MCB is a viable geoengineering scheme is mainly based on the assumption that the strong radiative forcing due to -Nd (IMO2020) can be mirrored to a +Nd scenario (e.g., MCB), again, relying on the assumption of a linear system. We know, however, aerosol indirect effects tend to saturate at high Nd (where already bright clouds cannot be brightened anymore). Given that we are still in a polluted (high-Nd) world, the actual cooling generated by +Nd might be much smaller than the warming from cleaning up the same amount Nd. For these reasons, I feel the authors need to justify their main conclusion in the context of nonlinearity.

Response: The point about the non-linearity is correct. In our previous paper, we highlighted this point (Yuan et al., 2023) as the two references listed by the reviewer. We note that the non-linearity is included in our calculation because the CF and LWP adjustments depend on several variables and can be non-linear. We added this point in the method section and the Gryspeerdt et al. (2023) reference. We note that after the clean-up by the IMO 2020, the background state is cleaner than previously, so we expect stronger forcing for the more pristine current conditions from a MCB experiment that would take place in the near future, and this is reflected in our calculations.

Some minor notes:

- - *pp. 2, paragraph 3, line 2, I wouldn't call shipping emission as "long-running inadvertent MCB experiment," as the composition of the ship exhaust can be quite different from what the community has proposed to use in the context of MCB (i.e., sea salt particles).*

Response: We agree with the reviewer's point regarding the aerosol composition. However, we think that does not make this analogy less valid in terms of creating a forcing from MCB.

- - *pp. 2, paragraph 3, line 6, "... by dimming clouds" I believe you meant dimming low clouds, not all clouds, I would be more precise here.*

Response: Corrected.

- - *Figure 1, I recommend adding some details to the caption, e.g., this is dN_d averaged over the year of 2020 (right?). On c), is $d\ln(N_d)$ actually shown, the color bar confuses me a little, is the maximum change of N_d only $\sim 3/cc$?*

Response: Modified. Yes, the maximum is 3/cc in the N_d change. We did not show $d\ln(N_d)$ here. Panel d shows the ratio of dN_d changes induced by IMO2020 and all human activities from pre-industrial conditions.

- - *pp. 3, third to the last line, how is ref. 16 relevant to this sentence?*

Response: Should be reference 21. Corrected.

- - *pp. 4, third to the last line, what does SOM represent?*

Response: SOM is supporting online material. Fixed.

- - *Figure 2, caption, "... forcing from cloud dimming," not all components are dimming the clouds, I see a significant portion of LWP adjustments actually contributes to brightening. Could you also indicate in Methods how you quantify the Twomey and LWP contributions separately based on Eqn. 3.*

Response: Changed. These are described in references 16 and 21.

- - *pp. 5, 4 lines above the figure, “likely” or “possibly”? Are there observational evidence of low-cloud albedo decreases in the North Atlantic since 2020, e.g., in satellite record (CERES)?*

Response: Likely. We have not looked at the cloud albedo data, but CERES cloud radiative effect data were lower since 2020.

- - *Figure 3 & 4, font is inconsistent with Figure 1 & 2 and looks a bit small. Figure 4 caption, “Details in Methods section” I did not see a relevant introduction of the CERES data in Methods.*

Response: Changed. CERES data introduction is added into the methods section.

- - *Pp. 6, line 1-2, I think reference(s) suggesting that interhemispheric contrast in forcing leads to substantial changes in monsoons and precipitation patterns is needed here.*
- - *Pp. 6, forth to the last line, “... in the contrast instead of ...” please check the wording.*

Response: Changed.

- - *Pp. 6, paragraph 2, on dN_d uncertainty, is it cost-efficient to run a GOCART ensemble to estimate this uncertainty? just curious...*

Response: Here we use replay with fixed meteorology. We explored a bit of the N_d uncertainty by using different emission scenarios as documented in the method section. We tested with 2021 data and the results are similar.

- - *I see Supporting Materials attached to the end, but they are not referenced at all in the main manuscript. Is this on purpose or something is missing?*

Response: The supporting materials are referenced as “SOM” throughout the text which is now explicitly defined.

References

Gryspeerd, E., Povey, A. C., Grainger, R. G., Hasekamp, O., Hsu, N. C., Mulcahy, J. P., Sayer, A. M., and Sorooshian, A.: Uncertainty in aerosol–cloud radiative forcing is driven by clean conditions, Atmos. Chem. Phys., 23, 4115–4122, <https://doi.org/10.5194/acp-23-4115-2023>, 2023.

Jia, H., Quaas, J. Nonlinearity of the cloud response postpones climate penalty of mitigating air pollution in polluted regions. Nat. Clim. Chang. 13, 943–950 (2023). <https://doi.org/10.1038/s41558-023-01775-5>

Decision letter and referee reports: second round

12th Apr 24

Dear Dr Yuan,

Your manuscript titled "Termination Shock of an Inadvertent Geoengineering Experiment Produces Substantial Radiative Warming" has now been seen by our reviewers, whose comments appear below. In light of their advice we are delighted to say that we are happy, in principle, to publish a suitably revised version in Communications Earth & Environment, if you can:

- (1) transparently discuss caveats and uncertainties (especially where claims are based on extrapolation) and moderate your conclusions to reflect the degree of confidence
- (2) add observational evidence from satellite data where possible, and
- (3) fully resolve the issues raised by reviewers 3 (comment 4) and 4 (second critical comment).

I appropriate, we will publish your article under the open access CC BY license (Creative Commons Attribution v4.0 International License).

We invite you to revise your paper one last time to address the remaining concerns of our reviewers. At the same time we ask that you edit your manuscript to comply with our format requirements and to maximise the accessibility and therefore the impact of your work.

EDITORIAL REQUESTS:

*****Please take care to match our formatting and policy requirements. We will check revised manuscript and return manuscripts that do not comply. Such requests will lead to delays. *****

SUBMISSION INFORMATION:

OPEN ACCESS:

Communications Earth & Environment is a fully open access journal. Articles are made freely accessible on publication under a CC BY license (Creative Commons Attribution 4.0 International License). This license allows maximum dissemination and re-use of open access materials and is preferred by many research funding bodies.

For further information about article processing charges, open access funding, and advice and support from Nature Research, please visit <https://www.nature.com/commsenv/article-processing-charges>

At acceptance, you will be provided with instructions for completing this CC BY license on behalf of all authors. This grants us the necessary permissions to publish your paper. Additionally, you will be asked to declare that all required third party permissions have been obtained, and to provide billing information in order to pay the article-processing charge (APC).

[Link Redacted]

Best regards,

Heike Langenberg, PhD
Chief Editor
Communications Earth & Environment

On X(Twitter): @CommsEarth

REVIEWERS' COMMENTS:

Reviewer #1 (Remarks to the Author):

The authors have reasonably addressed my previous comments and made necessary revisions to the manuscript. I have no further comments.

Reviewer #3 (Remarks to the Author):

I appreciate the efforts the authors made to improve the manuscript and to address my comments and concerns. Most of my concerns are sufficiently addressed. I think the manuscript is ready for publication.

Remaining minor points:

1. Regarding the scaling of modeled dNd with observed Nd, it makes sense to me if the systematic bias in Nd between model and obs is a scaling factor rather than an offset. Could you confirm this is actually the case (perhaps a scatter plot)?
2. Still missing dNd unit on Fig. 1.
3. Why not provide the uncertainty range of +/- 0.11 W/m² in the abstract?

This won't affect the fact that this manuscript is in its publishable form, but I just want to make this final point, and whether to consider it is up to the authors and the editor.

4. Regarding the point I made on the generalization of ship-track derived susceptibility to all marine warm clouds, I understand that it has been mentioned and discussed explicitly in Yuan et al. (2023), and the current manuscript refers to it and other previous studies when discussing the limitations of this generalization. My point is that for a high-profile paper like this one which is about to be published in one of the Nature's journals, the authors have the obligation to make the caveats/limitations of their study explicit for the audience in the main text. Again, this is just my personal feeling and I leave the decision for the authors and the editor to make. The reasons I think these limitations are important to acknowledge are:

a. According to the lead author's 2022 publication in Science Advances, the occurrence of ship tracks is at most a few percent, and there are a huge percentage of seeded conditions where the clouds are not brightened (at least from a satellite view).

b. According to the dNd map shown in this study, there are relatively large Nd changes in regions/regimes where ship tracks are rarely detected. Therefore, a large percentage of the current forcing estimate is from extrapolation/generalization.

c. Essentially, ship tracks are manifestation of “cloud brightening” to aerosol seeding, yet, studies (already cited in the current manuscript) have shown that there are conditions where marine low clouds show cloud darkening/dimming potentials to seeding, whose contribution is left out, by design, in the forcing estimate of this work.

d. Regarding the authors’ response: “Essentially, we show that detected ship-track samples cover broad range of conditions (including stability conditions captured by measures such as estimated inversion strength and SST etc.) and the assumption is that under similar conditions, aerosols have similar impact, which is a reasonable assumption in our opinion and stated in the text.” I have to say I disagree with this statement. According to what the lead author shown in Yuan et al. (2023), the ship-track samples indeed cover a broad range of conditions, but only when variables are considered individually (i.e. one at a time). It hasn’t been shown that ship-track samples cover a broad range of MET-variable combinations, which, to me, is what really matters, as one may find certain combinations of METs that lead to cloud dimming potential where shiptrack samples may cover the range of individual variable but not the combination.

Reviewer #4 (Remarks to the Author):

In this study, Yuan and Coauthors attempt to demonstrate that the change to low sulfur fuel in global shipping (IMO2020) has resulted in changes to marine low clouds that have had a measurable impact on reflected sunlight and the warming of the Earth. This is an extraordinary claim, and it requires extraordinarily strong evidence to be convincing. In my opinion, the authors have not provided such evidence.

The premise of the paper is very much in line with the lead authors recent papers that have demonstrated convincingly how aerosol from ship exhaust influences marine low clouds to cause them to be more reflective than non-polluted clouds. However, the lack of observational support in this manuscript is a significant departure from the previous papers, and I think that overwhelming observational evidence is needed to prove the premise of the paper.

I have two critical issues with the manuscript. The first is methodological. IMO2020 is supposed to result in regional changes to marine low clouds that have lower cloud droplet number concentrations (Nd) that have a lower albedo. There is also reason to suspect that cloud fraction (CF) will decrease as cloud condensation nuclei (CCN) from ship exhaust is reduced. For these changes to have a global energetic impact, their regional signatures should be easily diagnosed from actual data such as MODIS, VIIRS, or other satellite products without having to resort to the methods used in this paper such as the simulated results in Figure 2. Indeed, several of the leading developers of cloud property retrievals applied to MODIS and VIIRS are co-authors on this paper. I do not understand why increases to effective radius and decreases to optical depth between 2018 and 2022 in the main shipping lanes of the Atlantic, for instance, are not a prominent part of the evidence presented. Instead, the authors resort to a modeling approach using an AI algorithm to predict Nd. I find the AI method interesting but poorly validated. It is unclear the extent to which the AI algorithm to predict Nd would represent reality since no actual cloud data is shown to support it.

My second critical comment has to do with the energy budget discussions. Unless I’m misreading the intent, the authors are arguing that IMO2020 has had a measurable impact on net radiation (Figure 4 – there are two figure 4’s in my copy). Since the IMO2020 change happened in 2020, the trend in net solar radiation shows no significant change to the decadal trend at that point yet the authors argue that the cloud effect can account for a large fraction of the change in net radiation. The net radiation

has been increasing through the 2010's and continues into the 2020's with significant interannual variability, the cause of which is still being studied. The authors argue that the cloud change due to IMO2020 can explain 80% of the observed warming. I consider this as evidence that the effect they diagnose is likely much too large. If the author's claims are true, then the slope should have a significant change in 2020 beyond interannual variability which it does not.

Author Responses: second round

We appreciate reviewers' comments and suggestions. By addressing them, we believe that the quality of our manuscript is improved. We detail our responses in the following.

Reviewer #1 (Remarks to the Author):

The authors have reasonably addressed my previous comments and made necessary revisions to the manuscript. I have no further comments.

Response: Thank you!

Reviewer #3 (Remarks to the Author):

I appreciate the efforts the authors made to improve the manuscript and to address my comments and concerns. Most of my concerns are sufficiently addressed. I think the manuscript is ready for publication.

Remaining minor points:

1. Regarding the scaling of modeled dNd with observed Nd, it makes sense to me if the systematic bias in Nd between model and obs is a scaling factor rather than an offset. Could you confirm this is actually the case (perhaps a scatter plot)?

Response: Here's what the scatter plot looks like.

Figure r1: MODIS Nd vs GEOS Nd.

2. Still missing dNd unit on Fig. 1.

Response: Changed.

3. Why not provide the uncertainty range of +/- 0.11 W/m2 in the abstract?

Response: Thanks for the suggestion. Added.

This won't affect the fact that this manuscript is in its publishable form, but I just want to make this final point, and whether to consider it is up to the authors and the editor.

4. Regarding the point I made on the generalization of ship-track derived susceptibility to all marine warm clouds, I understand that it has been mentioned and discussed explicitly in Yuan et al. (2023), and the current manuscript refers to it and other previous studies when discussing the limitations of this generalization. My point is that for a high-profile paper like this one which is about to be published in one of the Nature's journals, the authors have the obligation to make the caveats/limitations of their study explicit for the audience in the main text. Again, this is just my personal feeling and I leave the decision for the authors and the editor to make. The reasons I think these limitations are important to acknowledge are:

a. According to the lead author's 2022 publication in Science Advances, the occurrence of ship tracks is at most a few percent, and there are a huge percentage of seeded conditions where the clouds are not brightened (at least from a satellite view).

b. According to the dNd map shown in this study, there are relatively large Nd changes in regions/regimes where ship tracks are rarely detected. Therefore, a large percentage of the current forcing estimate is from extrapolation/generalization.

c. Essentially, ship tracks are manifestation of "cloud brightening" to aerosol seeding, yet, studies (already cited in the current manuscript) have shown that there are conditions where marine low clouds show cloud darkening/dimming potentials to seeding, whose contribution is left out, by design, in the forcing estimate of this work.

d. Regarding the authors' response: "Essentially, we show that detected ship-track samples cover broad range of conditions (including stability conditions captured by measures such as estimated inversion strength and SST etc.) and the assumption is that under similar conditions, aerosols have similar impact, which is a reasonable assumption in our opinion and stated in the text." I have to say I disagree with this statement. According to what the lead author shown in Yuan et al. (2023), the ship-track samples indeed cover a broad range of conditions, but only when variables are considered individually (i.e. one at a time). It hasn't been shown that ship-track samples cover a broad range of MET-variable combinations, which, to me, is what really matters, as one may find certain combinations of METs that lead to cloud dimming potential where shiptrack samples may cover the range of individual variable but not the combination.

Response: Good idea. We have reiterated the assumptions and potential sources uncertainty of our method in the method section.

In regard to the four points the reviewer raised here, we want to briefly clarify.

a) While it is true that the percentage of tracks that are not detected is high, it is not necessarily the case that the physics are different. There are many possible causes that make these tracks not detectable. Our assumption is that the physics do not change.

b) The forcing is still mostly from regions that we have detected ship-track samples. There is extrapolation as noted in our 2023 paper, but their overall contribution to the forcing is small, roughly 10% or less.

c) these conditions are included in our samples. E.g., those cases with LWP reductions.

d) This is plausible, and we agree that future work is needed to address this question.

Reviewer #4 (Remarks to the Author):

In this study, Yuan and Coauthors attempt to demonstrate that the change to low sulfur fuel in global shipping (IMO2020) has resulted in changes to marine low clouds that have had a measurable impact on reflected sunlight and the warming of the Earth. This is an extraordinary claim, and it requires extraordinarily strong evidence to be convincing. In my opinion, the authors have not provided such evidence.

The premise of the paper is very much in line with the lead authors recent papers that have demonstrated convincingly how aerosol from ship exhaust influences marine low clouds to cause them to be more reflective than non-polluted clouds. However, the lack of observational support in this manuscript is a significant departure from the previous papers, and I think that overwhelming observational evidence is needed to prove the premise of the paper.

I have two critical issues with the manuscript. The first is methodological. IMO2020 is supposed to result in regional changes to marine low clouds that have lower cloud droplet number concentrations (Nd) that have a lower albedo. There is also reason to suspect that cloud fraction (CF) will decrease as cloud condensation nuclei (CCN) from ship exhaust is reduced. For these changes to have a global energetic impact, their regional signatures should be easily diagnosed from actual data such as MODIS, VIIRS, or other satellite products without having to resort to the methods used in this paper such as the simulated results in Figure 2. Indeed, several of the leading developers of cloud property retrievals applied to MODIS and VIIRS are co-authors on this paper. I do not understand why increases to effective radius and decreases to optical depth between 2018 and 2022 in the main shipping lanes of the Atlantic, for instance, are not a prominent part of the evidence presented. Instead, the authors resort to a modeling approach using an AI algorithm to predict Nd. I find the AI method interesting but poorly validated. It is unclear the extent to which the AI algorithm to predict Nd would represent reality since no actual cloud data is shown to support it.

Response: Thanks for raising the point. Indeed, changes in cloud properties over the southeast Atlantic due to IMO2020 have been shown in another study (Diamond, 2023) that is referenced in the manuscript. The attribution to ship emissions, however, requires a more sophisticated statistical technique. We nevertheless try to analyze the data based on this comment and show that observations from MODIS do show decrease in Nd and increase Reff after IMO2020, which is consistent with the reduction of emissions. This figure shows the Nd and Reff distributions over this region for the three years before and after IMO2020. Nd shows clear reduction along the shipping lane while Reff increases, which is consistent with Diamond (2023).

Fig r2: Three-year mean of Nd and Reff before and after 2020 for the SON season. The season is picked because signals from shipping emission on clouds are best observed at this time according to Diamond (2023). Their difference is shown in the bottom row.

My second critical comment has to do with the energy budget discussions. Unless I'm misreading the intent, the authors are arguing that IMO2020 has had a measurable impact on net radiation (Figure 4 – there are two figure 4's in my copy). Since the IMO2020 change happened in 2020, the trend in net solar radiation shows no significant change to the decadal trend at that point yet the authors argue that the cloud effect can account for a large fraction of the change in net radiation. The net radiation has been increasing through the 2010's and continues into the 2020's with significant interannual variability, the cause of which is still being studied. The authors argue that the cloud change due to IMO2020 can explain 80% of the observed warming. I consider this as evidence that the effect they diagnose is likely much too large. If the author's claims are true, then the slope should have a significant change in 2020 beyond interannual variability which it does not.

Response: We appreciate the reviewer's point/suggestion and carried out additional calculations to test it. First, we want to note that the forcing is a shock and it has been only 4 years since 2020. The energy imbalance measured by CERES has significant interannual fluctuations at interannual time scales. The short time period and the interannual variability do not yet permit robustly attributing the perturbation in the trend to the IMO2020 effect. The impact would be better detected after another decade or so. Nevertheless, we calculated the long-term trend at $0.46 \text{ Wm}^{-2}/\text{decade}$ while the trend since 2020 is $0.67 \text{ Wm}^{-2}/\text{decade}$. The difference is 0.21 Wm^{-2} , which is consistent with our forcing estimate. We revised the figure to reflect this in the text. We added discussion in light of the

reviewer’s suggestion. “The long-term trend of CERES TOA net radiation is $0.46\text{Wm}^{-2}/\text{decade}$ while it changes to $0.67\text{Wm}^{-2}/\text{decade}$ since IMO2020 took effect. The difference is 0.21Wm^{-2} that is consistent with our estimated forcing. However, the record is too short to ascertain the impact of IMO2020 on the long-term trend of the energy balance given its large interannual variations” to the last paragraph on page 6 reflect this point.

Fig r3: TOA net radiation time series with added long-term and post-2020 trends added.